# Systematic Review and Meta-Analysis of Transcendental Meditation for Post-Traumatic Stress Disorder

**DOI:** 10.3390/medicina61040659

**Published:** 2025-04-03

**Authors:** David W. Orme-Johnson, Vernon A. Barnes, Brian Rees, Jean Tobin

**Affiliations:** 1Faculty Emeritus, Maharishi International University, Fairfield, IA 52556, USA; 2Faculty Emeritus, Georgia Prevention Institute, Department of Medicine, Augusta University, Augusta, GA 30912, USA; vbarnes@augusta.edu; 3U.S. Army Reserve, San Luis Obispo, CA 93405, USA; brees@tm.org; 4Transcendental Meditation for Women, Maharishi Vedic City, IA 52556, USA; jtobin@tm-women.org

**Keywords:** PTSD, stress reduction, prolonged exposure therapy, facilitated support groups, meditation, Transcendental Meditation

## Abstract

*Background and Objectives.* Our recent systematic review and meta-analysis of all studies on meditation as treatment for PTSD (61 studies) found a moderate effect size of Hedges’s g = −0.67 for post-minus-pre change in symptom scores. Separate tests of the four meditation categories found a large effect size of g = −1.13 for the Transcendental Meditation (TM) technique that is significantly greater than for each other category. The present follow-up used a different method, calculating effects relative to internal controls, to better characterize the effects of this meditation technique. *Materials and Methods*. Our study followed Prisma guidelines. Major databases, research anthologies, and bibliographies were searched for studies that used TM for treating PTSD, all military and civilian populations, and all age groups. *Results.* The searches located 15 controlled trials on TM that met the inclusion criteria (longitudinal and reporting sufficient statistics to calculate effect sizes), 1248 subjects total, mean age 40.5 years (range 20.6 to 54.4 years), and 46.9% males (range 0% to 100%). Using the random effects model, the pooled effect across all studies of TM compared to other treatments was g = −1.01, 95% CI = −1.29 to −0.74, *p* < 0.000000001. One-study removed analysis found that no study reduced the pooled effect to less than −1.0. Funnel plots indicated no risk of bias. TM was non-inferior to prolonged exposure therapy, *p* = 0.0001, and it worked significantly faster (*p* = 0.04 at week six). *Conclusions.* TM produced clinically meaningful reductions in PTSD for civilian and military personnel, young and older adults, and for both men and women. We recommend phase-III multisite studies comparing TM with known first-line treatments for PTSD.

## 1. Introduction

The Problem. Every day, thousands of people are experiencing or witnessing terrifying and life-threatening events, causing post-traumatic stress disorder (PTSD) [1]. PTSD is a serious mental health condition that can include dissociative reactions (flashbacks) and nightmares; persistent avoidance of stimuli and memories associated with the trauma; uncontrollable intrusive thoughts and mood swings; and marked alterations in arousal, such as panic attacks, depression, and insomnia. People with PTSD may be hypervigilant, sleep poorly, distrust others, have memory problems, and have difficulty making decisions and following through. They are challenged both by outer circumstances and by inner conditions. Medical communities are under strain to treat PTSD, and the treatments available are only marginally effective [2,3,4].

PTSD affects approximately 6% to 7% of adults in the U.S. over their lifespan [5]. PTSD takes a severe toll on the mental, physical, and financial well-being of veterans and their families [6,7]. Conservatively, there are over a million veterans suffering from PTSD in the US today. It is estimated to affect 14% of those deployed in wars in Afghanistan and Iraq [8], and studies place lifetime prevalence for Vietnam Veterans as high as 30% [9]. Currently, 26.7% of Iraq and/or Afghanistan veterans who seek care from the Veterans Health Administration receive a PTSD diagnosis [5].

Limitations of Current Approaches. The two most recommended forms of treatment for PTSD use pharmaceuticals and/or forms of trauma-focused cognitive behavioral therapy, such as exposure therapy. Drugs can be targeted toward particular symptoms, such as depression, anxiety, insomnia, nightmares, etc., but it is generally recognized that there is no drug or drug cocktail that is specific for or can effectively cure, PTSD [6].

In a 35-year review of randomized controlled trials (RCTs) from 1980 to 2015, researchers found that the two gold standard treatments were lacking in effectiveness. Even after completing treatment, two-thirds of veterans with PTSD treated with prolonged exposure (PE) or cognitive processing therapy (CPT) retained their diagnosis [8]. A number of other studies looking more at ‘real world’ effectiveness have shown that many veterans find the ‘best’ treatments intolerably uncomfortable; completion rates can be extremely low, even 10% or less [10,11].

Meditation for Treating PTSD. A variety of essentially non-trauma-focused meditation practices have been considered in the treatment of PTSD [12,13,14,15,16]. In our PROSPERO protocol [17], we published our plan to conduct an updated systematic review and meta-analysis of all treatments identified as “meditation”, all research designs, and all populations. We presented the within-group results, which compared the change from pretest to posttest produced by each of four meditation categories. We located 61 studies with 3440 subjects total on Mindfulness-Based Stress Reduction (MBSR, 13 studies); Mindfulness-Based Other Techniques (MBO, 16 studies); Transcendental Meditation (TM, 18 studies); and Other Meditations that were neither mindfulness nor TM (OM, 14 studies). The baseline characteristics of subjects were similar across the meditation categories: mean age = 52.2 years, sample size = 55.4%, males = 65.1%, maximum study duration = 13.2 weeks. We found that many different populations with PTSD are willing to try and practice meditation: men and women, young adults and the elderly, war veterans, survivors of Military Sexual Trauma (MST), war refugees, earthquake survivors, prison inmates, female survivors of interpersonal violence, and clinical nurses. No study reported serious side effects. There were no significant differences between meditation categories in strength of research design nor evidence of publication bias. Nor were there appreciable differences in types of study populations included, outcome measures, control conditions, gender, or length of the studies.

All meditation techniques had statistically significant effects on reducing PTSD. The strength of the effects in Hedges’s g were moderate for MBSR (−0.52), MBO (−0.66), and OM (−0.63) and large for TM (−1.13). TM’s effect was significantly larger than for each of the other categories, which did not differ from each other. Meta-regression found that type of meditation was the most significant covariate, accounting for 41% of the variance (R^2^ = 0.41), with age accounting for 35% and trauma groups 26%. Multivariate meta-regression that included these three covariates taken together increased the R^2^ to 0.64, 64% of the variance explained by the covariates.

Having compared four different meditation techniques in our first paper, the objective of this second systematic review and meta-analysis is to examine the between-group effects of TM, which compares TM with various control groups on the study level. The between-group analyses comparing TM with control groups are designed to answer such questions as: Can the effects of TM on PTSD be explained by psychological variables (placebos). Is there an effect from the extra attention subjects get from just being in the study? Are there expectations of benefits they have from knowing which study group they are assigned? Another set of research questions addressed by these between-group studies revolves around how TM compares with well-known current best practices for treating PTSD, such as prolonged exposure or group psychotherapy. Is TM non-inferior to prolonged exposure or group psychotherapy with a professional facilitator?

In Part I of this paper, we present each study individually, describing the background of the problem of PTSD in that population, where the study was performed, who did it, and present the results and limitations with suggestions for future research that the study authors noted. In Part II we present results of our meta-analysis and meta-regression, including summary effect sizes, 95% confidence intervals, funnel plots and statistics, bias analyses, and the time course of the effects. Here we also present meta-regression findings on covariates that may potentially influence the results, such as study population, type of trauma, and the prediction interval (which measures heterogeneity, the range of effect sizes that future studies would be expected to find). We also report the proportion of variance not explained by TM and the control groups or the co-variates (I^2^) and propose future research.

## 2. Methods

Searches. Following Preferred Reporting Items for Systematic Reviews and Meta-Analyses guidelines (Prisma, https://www.prisma.io/ accessed on 15 March 2025), we searched major databases (PubMed, MEDLINE, PsycINFO, Web of Science, Library of Congress, and Google Scholar), as well as the bibliographies of systematic reviews and meta-analyses, bibliographies of original research papers, and research anthologies and databases of meditation research, from January 1970 through June 2024, for papers in English, published and unpublished, on Transcendental Meditation, TM, and PTSD. We searched the bibliographies of major reviews and meta-analyses on meditation research.

Types of Studies Included. We included controlled longitudinal research designs on TM that measure change over time that have sufficient data to enable calculation of effect sizes and variances at each measurement period. This includes randomized controlled trials (RCTs) in which participants are randomized to either the treatment or control/comparison group(s). It also includes controlled trials (CTs), which are studies similar to RCTs, except that the participants self-select to be in the treatment group, or they are assigned to groups by some non-random procedure.

Participants/Population. Inclusion criteria were populations at least presumptively diagnosed with PTSD that had TM in at least one arm of the study, all age groups, and all gender identifications. Populations included combat veterans, war refugees, male and female prison inmates, college students, urban trauma survivors, and nurses under stress.

Studies were excluded if they did not have sufficient data for calculating effect sizes, were not longitudinal with a pretest and at least one post-test, were correlational and did not measure treatment effects, were reviews only, or were studies where we did not have permission to use their data.

Data Coding. Full copies of the qualified papers were obtained, and their statistical data, including age, gender, duration of the treatment, and PTSD measures, were extracted and entered into the Comprehensive Meta-Analysis Program (CMA) [18]. Because long-term outcomes are most clinically meaningful, we used the effect sizes from the most recent available follow-up data.

Dependent Variables. The primary outcome measure is total PTSD symptoms, measured by the Abbreviated PTSD Checklist, PCL, a 20-item questionnaire, corresponding to the DSM-5 symptom criteria for PTSD [19] or the older DSM-4 based 17-item PCL [20]. The PCL is the most widely used measure of PTSD. We used the Hedge’s g statistic, which expresses the difference of the experimental mean minus the control mean in units of the pooled standard deviation, adjusted for sample sizes [21]. Some studies also used the Clinician-Administered PTSD Scale (CAPS), a semi-structured interview, for determining PTSD diagnosis and symptom severity [22]. The CAPS is considered the “gold standard” in the field. It provides Cohen’s d. When only Cohen’s d was reported, in order to compare all studies on the same metric, we converted d into g, where the conversion of d to g is g = d times a correction factor J: J = 1 − 3/(4 df − 1) [23].

For studies that had repeated measures, we reported the time course of the change in PTSD symptoms. These data were reported in the papers: Bandy et al., 2019 [24], Rees et al., 2013 [25], Bonamer et al., 2024 [26], and Leach and Lorenzon, 2023 [27]. For the Nidich et al., 2018 [28] study, we obtained data on the reduction in the PCL at the various time points from one of the co-authors and the statistician on the paper, Dr. Maxwell Rainforth. Three studies did not report results with PCL or CAPS. For plotting change scores in PCL points, we converted the g’s for these studies to PCL points. We obtained full-text PDFs of the 15 included studies, and there was no missing data in them.

Calculating Effect Size. Our methods followed the Introduction to Meta-Analysis [23]. We used the Comprehensive Meta-Analysis (CMA) program for the core analysis of effect sizes, bias analysis, and meta-regression [18]. The random-effects model was used because of the wide differences between study populations on initial levels of anxiety, age, and other variables. The core analysis included the point estimate of the standard difference in the means, Hedges’s g’s, including standard errors, variances, 95% confidence intervals (95% CI), Z scores, statistical significances, and Forest plots. Hedges’ g corrects the standardized difference in means (Cohen’s d) by applying a factor “J” because it accounts for potential bias introduced by small sample sizes in individual studies, essentially providing a more accurate estimate of the true effect size when dealing with small sample groups. This correction factor adjusts the calculated Cohen’s d to be less inflated in such situations. The CMA program first calculated the standardized mean difference (d) and then multiplied d by a correction factor (J) to compute g. The standardized mean difference (d) was standardized by the pooled Post score SD. The formula for the pooled Post SD was:

SDPost(1) = Given, 

SDPost(2) = Given,

SDPostPooled = Sqr(((n(1) − 1) × SDPost(1)^2^ + (n(2) − 1) × SDPost(2)^2^)/(n(1) + n(2) − 2)).

This yields the effect in Cohen’s d, which was converted to Hedges’s g by multiplying it by the correction factor J, J = 1 − (3/(4 × df − 1)) [23].

Heterogeneity was assessed by using the Q statistic and its associated *p*-value and the I^2^. I^2^ indexes heterogeneity by giving the proportion of the observed variance that is real between-studies variance, in contrast to variance due to sampling error of a common underlying effect (fixed effect).

Outlier. The Rees et al., 2013 [25] study was an outlier, with an ES of g = −8.273, which was almost 8 times the pooled ES of all 15 studies taken together, g = −1.208. Removing Rees et al., 2013 [25] only reduced the pooled effect from −1.208 to −1.01. One-study removed analysis found that removal of this study resulted in a point value of −0.9895, and the point-value of removing any of the other studies was over −1.0. (See Appendix A for details).

Synthesis Methods. We used different equivalent modes of data entry available on the Comprehensive Meta-Analysis program (CMA) [29] to accommodate different methods of reporting data by the individual papers. These included data reported as the means, differences in the means, mean changes, standard deviations, variances, Ns, *p*-values, correlations, 95% confidence intervals, and pre-post correlations. Whatever statistics were required were entered into the appropriate modules of the CMA to create tables of summary effects, 95% CIs, I-squared statistics, tau, and tau squared, as well as multiple regression analyses. We used the Microsoft graphics packages Excel, PowerPoint, and Word to create bar charts and line graphs. We used the standard CMA graphics modules to create forest plot and the scatter diagrams. We obtained full-text PDFs of the 15 included studies, and there was no missing data in the studies.

Three-Armed Studies. A three-armed study is one that has three groups being compared, such as TM being compared with both prolonged exposure (PE) and health education (HE). Two other such studies compared TM, Adapted Mantra Meditation, and Patient Centered Therapy, and compared TM’s effects on two earthquake trauma groups (at Sendai and Ishinomaki, Japan) compared to one non-trauma TM group. The statistical issue in such studies is that meta-analysis assumes statistical independence of the studies. The solution in such cases is to enter half of the N for the study that is being used twice [23]. For example, if the TM group has 68 subjects, and it is compared with two control groups, then 34 (half of the n) is entered for the TM group in the between-group comparisons of TM with each of the two controls. This ensures there is no duplication of the study weight in the overall synthesis, although the statistical power of these two comparisons is somewhat reduced. 

Bias Analysis. Publication bias was assessed by inspection of funnel diagrams, and their asymmetries were quantified by the Begg and Mazumdar rank correlation and Egger regression intercept tests [23]. The Rosenthal classic fail-safe N was used to calculate how many missing non-significant studies it would take to reduce the mean effect size to non-significance [30].

## 3. Results

Literature Search. This paper is the second of two systematic reviews and meta-analysis of the effects of meditation in the treatment of PTSD. A complete description of the literature search with a flow diagram is in the first paper [31]. That search included all research designs, RCTs, CTs, and Single Group studies. We located 18 RCT, CT, and single group studies that used the TM technique for treating PTSD. Of those, 15 were RCTs or CTs. The present paper reports the between-group results of these 15 controlled TM studies, the controlled TM studies, RCTs, and CTs in both military and civilian populations. The flow diagram and description of the search can be found in Appendix A Flow Diagram and Description of Literature Search.

We present the results in two sections. Part I of the results section presents the study-level results of each paper as bar charts using data from the papers comparing TM with various control groups. The statistical analyses used and the results are those reported in the papers. Part II of the results presents the meta-analytical synthesis of all the studies together and the meta-regression analyses.

### 3.1. Part 1: Between Group Analyses Comparing TM with Various Controls

#### 3.1.1. Individual Studies

Figure 1 shows the results for Brooks and Scarano [32], Heffner et al., 2014 [13], Nidich et al., 2018 [28], and Bellehsen et al., 2021 [33]. The vertical axis is the change in PTSD symptoms on the PCL-M plus or minus the SE. In all four studies, the TM group produced clinically meaningful reductions in PTSD symptoms, as indicated by a reduction of 10 or more points on the PCL-M.

Brooks and Scarano, 1985 [32]. This study was conducted at the Denver Vietnam Veterans Outreach Program. It compared 10 subjects treated with TM to 8 treated with psychotherapy, which included weekly individual psychotherapy sessions and group or family counseling. The psychotherapeutic treatments were eclectic and included behavioral, existential, cognitive, somatic, and psychodynamic treatment, according to the training of the therapist. Subjects were assigned every other person to the TM or psychotherapy, which, according to the Cochrane handbook, is quasi-randomization [34]. After 12 weeks, there were significant reductions in the TM group compared to psychotherapy on PTSD symptoms measured by a scale modeled after DMS III criteria. TM subjects also decreased compared to controls on subscales of emotional numbness and on co-morbid conditions of PTSD: depression, anxiety, alcohol consumption, and electrophysiological stress reactivity (skin resistance startle response to loud tone [35]), as well as improvements in sleep quality, family life, and employment problems. The authors commented (p. 214): “The TM program may sufficiently relieve the symptoms of many individuals with PTSD. In some cases, however, a combination of approaches of both TM and psychotherapy (or other approaches) may be the preferred treatment”.

Heffner et al., 2014 [13]. This study was a part of a multi-site demonstration project that studied different meditation techniques for treating PTSD and was commissioned by the Department of Veteran Affairs [13]. A three-armed RCT comparing the TM technique (N = 19) with Adapted Mantra Meditation (AMM, N = 22) and Present-Centered Therapy (PCT, N = 24) was conducted at the Aleda E. Lutz VA Medical Center, Saginaw, Michigan. Adapted Mantra Meditation (AMM) is a standardized group-based teaching method for Mantra Meditation described as a non-spiritual style of meditation with all members using the same generic mantra during their instruction and daily practice. Present-Centered Therapy (PCT) is a non-trauma focused treatment for PTSD originally developed as a strong comparator treatment that captured many of the effective components of “good psychotherapy” [36]. It includes: (1) psychoeducation on PTSD to help patients understand how symptoms are disrupting their day-to-day functioning; (2) effective strategies for approaching day-to-day challenges; and (3) homework outside the session by which individuals can monitor stressors and practice new problem-solving skills [37]. PCT does not include disclosure, discussion, or exposure to traumatic events and does not focus on cognitive restructuring or relaxation training. It is a structured treatment with homework assigned between sessions.

Inclusion criteria were active PTSD with intake PCL scores > 30 (a criterion of a PTSD diagnosis), no evidence of active substance abuse, no present or previous diagnosis of any disorder with psychotic features, commitment to completing the daily assignments, approval of current treatment providers, and agreement to refrain from other psychosocial treatments during the study. Qualifying patients were randomly assigned to one of the three possible treatments.

TM produced a greater reduction in PTSD compared to AMM and PTC. The significance of the group difference in change across time was *p* = 0.001 [13]. The study concluded that: “For both the clinician-administered and the self-administered measures of PTSD severity, veterans in the Adapted Mantra and Present-Centered Therapy programs showed a decline of a medium magnitude, while veterans in the Transcendental Meditation™ program showed statistically large declines” [13], p. 68. No adverse events were reported for any group. This replicated the results of Brooks and Scarano, 1985 [32], published 29 years previously.

Limitations noted were costly participation due to the proprietary program and TM’s traditional initiation ceremony, which raised concerns among a few Veterans. Future studies on the combined effects of TM and PCT together may find stronger effects than TM by itself.

Nidich et al., 2018 [28]. The third and highest quality RCT of TM on PTSD was a non-inferiority study comparing TM to Prolonged Exposure therapy (PE), a first-line treatment for PTSD used by the VA [28]. It was conducted at the Department of Veterans Affairs San Diego Healthcare System in California. The authors randomly assigned 203 veterans with a current diagnosis of PTSD resulting from active military service to TM (N = 68) or PE group (N = 68), or an active control group of Health Education for PTSD patients (HE, N = 67). Each treatment provided 12 sessions over 12 weeks, with daily home practice. TM and HE were mainly given in a group setting, and PE was given individually. The primary outcome was change in PTSD symptom severity over 3 months, assessed by the Clinician-Administered PTSD Scale (CAPS). Secondary outcomes were self-reported PTSD symptoms, using the PCL-M, and depression (PHQ-9).

The non-inferiority analysis comparing TM with Prolonged Exposure Therapy (PE) found that TM was non-inferior to PE (*p* = 0.0001) on the CAPS score and non-inferior to PE for the PCL-M and PHQ-9 scores (*p*’s = 0.0002) when covarying using baseline-dependent score, number of PTSD medications, gender, and number of years since discharge from the armed forces as covariates. Similar results were obtained when including the following additional covariates: antidepressants and antipsychotic medications at baseline, change in number of PTSD medications, baseline social support, and number of treatment sessions.

Superiority analysis found that TM was not statistically superior to PE at the final posttest (13 weeks). However, TM produced a 21% greater reduction in global PTSD symptoms than PE at posttest, as shown in Figure 1. Moreover, TM reduced PTSD symptoms faster than PE, creating a clinically meaningful reduction in four weeks that was statistically more effective than PE by week 6 (*p* = 0.04). (See the section below on the rate of change of treatments). 61% of those receiving TM, 42% of those receiving PE, and 32% of those receiving HE showed clinically significant improvements. There were no treatment-related adverse events for any of the three treatments.

Nidich et al., 2018 [28] further validated TM for treating PTSD by showing that it is at least as effective as a prolonged exposure treatment (PE). It is considered the highest quality study on TM for PTSD because it used well-established, validated randomization procedures, control groups that received a parallel treatment schedule provided by skilled therapists with systematic follow-up, and intent-to-treat analysis.

The authors noted four limitations in this study. (1) They found smaller effects for PE than previous studies. This was perhaps because of the limited experience of the therapists or high baseline levels of PTSD. (2) The study lacked sufficient follow-up regarding possible further reductions in PTSD and the assessment of the durability of PE and TM. (3) Treatment dropout was a potential confound because PE had a higher dropout rate than TM, although it was not statistically significant. (4) TM instructors had more experience teaching groups other than PE compared with other study therapists. Nevertheless, both TM and other study therapists were novices regarding delivering treatment to PTSD veterans. The study was unable to recruit enough women to enable TM and PE comparisons by gender. This remains an issue for future research to address.

Bellehsen et al., 2021 [33]. The fourth controlled trial of TM for treating PTSD in veterans was an RCT conducted at the Unified Behavioral Health Center for Military Veterans and their Families (UBHC), the Northport Veterans Administration Medical Center in Long Island, New York [33]. Veterans with PTSD (N = 40) were randomly assigned to a TM intervention or treatment-as-usual (TAU) control group, with 20 participants in each group. The inclusion criteria were (a) a diagnosis of PTSD, per the criteria in the DSM-5, made using the CAPS-5); (b) an elapsed time of 3 months or more since exposure to an index traumatic event; (c) a stable treatment regimen of medication and/or psychotherapy for PTSD that had been in place for at least 2 months before study enrollment; and (d) 18 years of age or older. 85% were male. Participants were excluded from the study if they (a) experienced current acute symptoms of psychosis or (b) reported substance dependence that had not been in remission for at least 3 months.

Change in PTSD symptoms, measured via the Clinician-Administered PTSD Scale for DSM-5 (CAPS-5), was the primary outcome. Secondary outcomes included the PCL, depression, anxiety, sleep difficulties, anger, and quality of life. Assessments were conducted at baseline and 3-month follow-up.

Figure 1 shows that for Bellehsen et al., 2021 [33], the TM group decreased by 11.28 points on the CAPS-5 compared to a 1.62-point decrease for the treatment-as-usual control (*p* < 0.006). At posttest, 50.0% of veterans in the TM group no longer met PTSD diagnostic criteria as compared to 10.0% in the TAU group, *p* = 0.007. Adjusted mean changes on self-report measures of depression, anxiety, and sleep difficulties indicated significant reductions in the TM group compared to TAU, d’s = −0.80 and −1.16. There were no significant group differences regarding anger or quality of life. No adverse events were reported.

“Weaknesses of this study included a lack of a time-and-attention control, exclusion of participants with co-morbidities such as substance dependence, and an absence of a follow-up period to assess longer-term stability of change as well as sustainment of meditation practice.” (p. 15). The authors conclude that the study demonstrates the efficacy of TM as a treatment for veterans with PTSD and for comorbid symptoms by a tolerable, non–trauma-focused PTSD treatment [32].

Figure 2 displays the results for four civilian groups. 

Yoshimura et al., 2015 [38], Earthquake Victims. On 11 March 2011, the largest earthquake in Japanese history, the great East Japan earthquake and tsunami, killed almost 20,000 people, destroyed 138,000 buildings, and caused over USD 200 billion in economic damage [39]. The damage of the earthquake and the tension caused by the Fukushima nuclear reactor difficulties were a great source of duress for the region and the nation. Many individuals in the cities affected by the disaster experienced trauma either directly from the disaster or indirectly through the psychological duress of caring for those who had lost family members.

The study by Yoshimura and colleagues (2015) [38] documented changes in self-reported stress symptoms after instruction in TM among 171 residents of two Japanese cities (Sendai and Ishinomaki), who were directly affected by the earthquake and tsunami disaster. The study occurred 2 to 5 months after the earthquake in Sendai and 5 to 8 months afterward in Ishinomaki. At pretest, subjects in both groups were still experiencing high levels of PTSD.

To test the hypothesis that the TM technique would quickly alleviate PTSD symptoms, TM subjects were compared with a no-treatment control group (n = 68) tested before and one week after learning the technique [38]. Figure 3 shows that one-week of TM practice produced large reductions in PTSD indicators in both Sendai and Ishinomaki residents compared to no-treatment controls. The TM groups had a 42.4% reduction in total symptoms from the pretest. The percent of subjects rating that “no symptoms were experienced always or very much” increased after one week of TM practice from 25% to 75% for Sendai and from 37% to 85% for Ishinomaki. PTSD symptoms for controls remained largely unchanged, increasing from 57% to 62%. No adverse events were reported.

Seventy-nine percent of the Japanese subjects were women. At pretest, women reported significantly more symptoms than men did (*p* = 0.001). In Japanese culture, women may feel more comfortable than men in reporting perceived problems. This was similar to the experience of offering TM for disaster relief from the earthquake in Soviet Armenia in 1989–1990. In these two very different cultures, Armenia and Japan, women were four times more likely to admit psychological duress than men and were more likely to come forth for psychological help than men [38,39]. However, in this study in Japan, as was our experience in Armenia, the effects of TM in reducing post-traumatic stress symptoms were equally evident for both men and women.

The rapid reported relief of PTSD raises the question of whether the relief would be lasting. Moreover, self-selection and the short treatment period did not rule out the contribution of motivation effects. The authors concluded: “These preliminary findings warrant replication in disaster settings using a randomized controlled trial design with an alternative treatment and with both short-term and long-term assessment with standard scales.” (p. 10).

Bandy et al., 2019 [24], Traumatized South African College Students. Crime, domestic violence, and sexual assault are high in South Africa, resulting in an 8% to 38% prevalence of PTSD symptoms in South Africa adolescents and children. Bandy et al., 2019 suggested that a widespread availability of the TM technique in these populations could potentially prevent and alleviate PT and depressive symptoms [24]. They instructed 34 students, two-thirds women, in the Transcendental Meditation technique at an experimental university in South Africa who were clinically diagnosed with PTSD and compared them with 34 students at a demographically comparable university who also were diagnosed with PTSD. Types of traumas experienced by the students included sexual and criminal victimization, natural disasters, severe accidents, and combat experiences.

Figure 2 shows a 21-point decrease on the PCL-C in the TM group after 15 weeks compared to no change in the control group. This large effect is considered clinically meaningful. The between-group effect was large (g = −2.28. *p* < 0.000001). After 15 weeks, the TM group, who started at a high level of PTSD (mean score of over 50), fell to a score of 32. No adverse events were reported.

The study found that TM practice produced significant reductions in PTSD after only 2 weeks. The rapid effects of TM in the first 15 days were significantly greater in students who regularly practiced TM twice a day as instructed (*p* = 0.009). Similar large decreases in symptoms were found for depression. The unusually large effect sizes in this study appear to be due in part to an effort by the university to ensure the regularity of practice of TM through twice-daily monitoring group meditations in class.

The authors noted the following limitations. “Several potential threats to validity were addressed and remedied. First, at pretest, the comparison group students had significantly higher PCL-C scores. This was accounted for statistically by covarying the pretest scores for both groups in the MANCOVA analysis. Second, two different clinicians diagnosed PTSD, one at each institution, but potential differences in clinician application of criteria should largely have been encompassed by the use of the pretest covariate.” (p. 10). Another potential confound is that the TM group practiced meditation on a regular twice-day basis as part of their regular university class schedule, whereas the control group did not have a technique to practice. This may have created motivation effects in the TM students that were not present in controls. What is needed are randomized controlled trials comparing TM with treatments practiced on parallel schedules with parallel motivational factors, such as a similar body of research showing beneficial effects and teachers with parallel experience in training PTSD patients who are at equivalent levels of PTSD severity.

Bonamer et al., 2024 [26], Clinical Nurses. The availability of a nursing workforce and the quality and safety of patient care depend on the health of clinical nurses. Nurses’ health is compromised by long hours, rotating shifts, and intense physically and emotionally challenging situations, including violence from patients and visitors. These stressors are well known to increase anxiety, depression, compassion fatigue, and burnout [26]. The COVID-19 pandemic further increased strain on healthcare providers, leading to reports of increases in the number of nurses with burnout and PTSD, resulting in alarming numbers of nurses reporting their intention to leave the profession [40], which would be a disaster for the nation’s health.

Bonamer and colleagues [26,41,42] offered TM to nurses as a tool for increasing multidimensional well-being conceptualized as the presence of flourishing and the absence of burnout, anxiety, and PTSD. One hundred and four nurses, 98 women and 6 men, in three hospitals in the Sarasota Florida Memorial Health Care System, were randomly assigned to either the TM group (N = 53) or the control group, who participated in a Group Support program (N = 49). Subjects were pretested and post-tested after 4 and 12 weeks on the PCL-5 C, the General Anxiety Disorder-7, the Maslach Burnout Inventory (MBI, considered the gold standard for evaluating burnout in healthcare providers), and the Secure Flourish Index of Well-being, which measures happiness/life satisfaction, physical/mental health, meaning/purpose, character, virtue, close personal relationships, and financial/material stability [40]. Of the nurses offered TM, 98% attended all 4 days of the initial training. Over 90% practiced TM at least once daily for at least 15 min most days of the week. The TM group improved compared to controls on well-being, including significant reductions of PTSD, anxiety, and burnout, and improvement in flourishing. Figure 2 shows large statistically and clinically significant reductions of PTSD symptoms in the TM group compared to the waitlist controls. No adverse events were reported.

The clinical nurses in this study typically work 12-h shifts and reported difficulty finding time to meditate twice a day on workdays. “With the potential benefit of TM practice to the health of the nurse and potentially their patients, organizations may find it desirable to not only support, but indeed encourage, space and time for TM practice within a 12-h shift. In addition, nurses may find it helpful to network with others within their work environments for group meditations planned immediately prior to or following their shifts. Benefits of group meditation have recently been described in the literature.” (p. 6 [25]).

Leach and Lorenzon, 2023 [27], Female survivors of Interpersonal Violence (IPV). Domestic violence is any physical, psychological, sexual, economic, or emotional action, or threat of action, that aims to maintain control and power over an intimate partner [43]. It can lead to serious injury, sexually transmitted infections, adverse pregnancy outcomes, sleep problems, hypertension, substance misuse, depression, anxiety, PTSD, homelessness, risk of suicide, and death. Globally, 26% of women (753 million) aged 15 years or older have experienced domestic violence at least once in their lifetime [44]. Best practice guidelines include relaxation techniques, counseling, cognitive behavioral therapy, and group work [27].

Leach and Lorenson, 2023 [27] studied the effects of the Transcendental Meditation technique and facilitated Group Support on 42 Female Survivors of IPV in Australia, one-third of whom had experienced trauma within the last year and two-thirds in more than a year previous to the study. The study found that, at the final posttest at 16 weeks, both treatments had significant effects on reducing Total PTSD symptoms (*p* = 0.027). At 16 weeks, the TM group produced a moderately larger effect than Group Therapy, which did not reach statistical significance (*g* = −0.43, *p* = 0.156, see Figure 2). The within-group change for TM was *g* = −0.769, *p* = 0.0001 compared to Group Support, which was *g* = −0.519, *p* = 0.002. However, TM reduced PTSD symptoms more rapidly than Group Therapy, as indicated by a significantly larger effect at week 8 (between-group comparison was *g* = −0.92, *p* = 0.004). The rate of change of different treatments will be addressed more fully in a later section.

Both TM and Group Support had a high level of adherence, 90% for TM and 88% for Group Support. For weeks 1–8, adherence to the twice a day TM schedule was 52.3% (approximately once a day) and 81.6% for weeks 9–16. Twelve adverse events reported by six TM group participants were transient and mostly mild (i.e., nausea, headache, irritability, and weight gain). Two participants self-reported a severe adverse event that they believed was related to the intervention (i.e., cold sore, body feeling heavy). Five adverse events reported by five participants in the control group were of mild-moderate severity and transient in nature and included shaking, feeling overwhelmed and upset, and having heart palpitations. The frequency of adverse events was not shown to be statistically significantly different between groups.

Twelve mild transient adverse events (i.e., nausea, headache, irritability, and weight gain) were self-reported by six of the TM participants. Two participants self-reported a severe adverse event that they believed was related to the intervention, i.e., a cold sore, (which is not likely due to meditation), and the body feeling heavy. In subjects participating in Group Support, five self-reported moderate to strong transient adverse events, including shaking, feeling overwhelmed, feeling upset, and having heart palpitations. This was likely due to them reliving traumatic events during group work.

Participants’ experiences in both TM and Group Support were reported as “overwhelmingly positive.” ([27] p. 6). TM subjects said that they were “grateful for the experience”, that it was “easy to learn”, that the instructors were “warm and accepting”, that the classes were enjoyable, and that TM “provided a lot of benefits”, including perceived improvements in mental clarity, eyesight, sleep, stress, and coping. Working the TM schedule of 20 min twice a day into one’s daily schedule is often challenging, but in this study, only two participants indicated this was a problem. Participants in the Support Group found that their discussions with other women were “empowering”.

Limitations. The study was not able to achieve its target sample size (88) specified by power analysis. However, post-hoc power analysis revealed that the 42 subjects it did recruit were adequate (power = 82%). Second, as is usually the case in behavioral studies, it was not possible to blind subjects to the intervention they received, which potentially could have introduced expectation bias. However, the likelihood of expectation bias is minimal because both arms of the study used active interventions and reported positive experiences with their interventions. Third, as participants were limited to women residing in an urban area (the Southern suburbs of Adelaide), it is not known whether the findings would be translatable to men or others living in non-metropolitan regions. These populations should be a focus of future research.

Studies on Civilian personnel: Male and Female Prison Inmates, Ukrainian and Congo War Refugees.

The studies shown in Figure 4 indicate that TM practice is effective for treating two of the most highly traumatized populations: prison inmates and war refugees.

**Figure 3 medicina-61-00659-f003:**
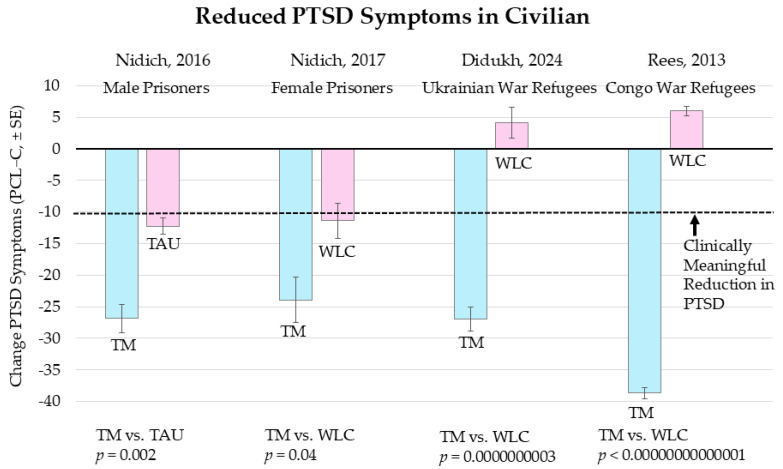
Results of controlled trials using the Transcendental Meditation technique for treating PTSD in: Male Prison Inmates, Nidich et al., 2016 [45]; Female Prison Inmates, Nidich et al., 2017 [46]; Ukrainian War Refugees, Didukh and Freytag, 2024 [47]; and Congo War Refugees, Rees et al., 2013 [25]. Abbreviations: TM = Transcendental Meditation, TAU = treatment as usual, WLC = waitlist control.

Nidich et al., 2016 [45], Male Prison Inmates. Prison inmates typically have a lifetime of trauma exposure, being raised and living in communities rampant with poverty, crime, interpersonal violence, racial tensions, and people making poor life-style choices [45]. For male prison inmates, TM practice had large effects on reducing total PTSD symptoms on the Total Trauma Symptoms Checklist [47]: 19.6 points, g = 2.15, SE = 0.180 for TM and (*p* < 0.001), which is 26.84 (2.25) PCL for TM and 12.25 (1.3) PCL points for the WLC group shown in Figure 3. TM also reduced the subscales of the Total Trauma Symptoms Checklist. For male inmates, the comparisons of the TM group with the TAU control group on the subscales were: dissociation (d = −0.79, *p* < 0.001), depression (d = −0.78, *p* < 0.001), anxiety (d = −0.67, *p* = 0.003), sleep disturbance subscales (d = −0.75, *p* = 0.001), and Perceived Stress (d = −0.89, *p* < 0.001) [45].

Use of no-treatment control was a limitation because some of the benefits associated with TM may not have been specific to TM, such as the extra attention they received from caring providers. Also, the four-month posttest limited the ability to assess long-term stability and possible continued improvements in trauma symptoms from continued practice of TM. “Future studies of trauma symptoms in prison populations should focus specifically on inmates with documented posttraumatic stress disorder and should take into account other psychiatric disorders that may be present, as well as standard psychotherapy and drug treatments being administered. Future research also should assess the relationship of trauma symptoms to functional impairment and other quality of life issues.” (p. 4).

Nidich et al., 2017 [46], Female Prison Inmates. The Nidich et al., 2017 [46] study with female prison inmates found significant decreases in Total Trauma on the PCL-C in the TM group compared with wait-list controls: total trauma score, d = −0.84, *p* = 0.036. PTSD subscales for intrusions (fewer repeated disturbing memories, thoughts, or images of a stressful experience; fewer disturbing dreams of a stressful experience from the past; d = −0.99, *p* = 0.026) and reduced hyperarousal (less feeling jumpy or easily startled; less being “super alert” or watchful on guard) (d = −0.82, *p* = 0.043) [48].

Didukh and Freytag, 2023, 2024 [47,48], Ukrainian War Refugees. Since the beginning of the Russian invasion, millions of Ukrainians have been exposed to life-threatening or profoundly disturbing events, and more than six and a half million have fled their country as refugees [49]. Living in foreign lands in temporary, improvised housing, disconnected from their families, jobs, and communities, daily hearing news of death and destruction in their homeland, many suffer from major depression, pervasive low mood, loss of interest or pleasure in almost all activities, pervasive fatigue, slowing down of thought, low self-esteem, and other symptoms of depression and PTSD. Consequently, there is an increased demand for treatments of PTSD and depression in Ukraine as well as in countries hosting Ukrainian refugees.

This study was conducted with 40 Ukrainian refugees in Lübeck, Northern Germany, practicing the TM technique compared to 40 control subjects who were interested in learning TM but did not practice it. All subjects were pretested and then post-tested at 30 days and 60 days using the Ukrainian versions of the PCL and Beck Depression scales [47].

Figure 3 shows that at 60 days there was a 27-point decrease in total PCL score in the TM group compared to a 4-point increase in controls, *p =* 0.0000000003, with similar changes in depression. The control group increased significantly in depression during the 30-to-60-day interval (*p* = 0.028), apparently due to having to remain living under refugee conditions without a technique to help increase their resilience. No adverse events were noted.

A study limitation is that the control group did not match the TM group in distribution of gender and prevalence of probable PTSD at baseline measurement. On all outcome measures, the control group tended to score slightly higher initially. Subjects were accepted to the study without a criterion for minimum symptom severity at baseline. However, the lower initial level of PTSD in TM subjects would have resulted in underestimating its effects.

Another possible confound was the housing situation. Some subjects lived in the local refugee hostels, whereas others lived in the private homes of German residents. It is not known if this advantaged one group or another. Moreover, although neither the control group nor the TM group received psychological treatment, the one-on-one check-up meetings of the TM group may have had a stabilizing effect.

Rees et al., 2013 [25], Congo War Refugees. The Second Congo War (1998–2003), also known as the Great African War, involved nine African countries and around twenty-five armed groups who killed 5.4 million people and forced an estimated 80,000 refugees to flee [50,51]. Rees and colleagues published two studies on the effects of TM on refugees from the Democratic Republic of the Congo [25,52]. They were staying in or around Kampala, Uganda, in temporary shelters, such as churches or rented accommodations. They were typically unemployed and had minimal access to mental health services. This population had been exposed to combat stress, sexual assault, torture, and/or forced to witness the abuse or killing of loved ones.

The first study by Rees and colleagues [25] compared 21 subjects who learned the TM technique immediately with 21 subjects who waited to learn TM until the end of the study after 135 days (19 weeks). Subjects were initially randomly assigned to groups, but the randomization was broken when a substantial number of subjects did not show up for TM instruction after pretesting. Participants were given a kilogram of beans and rice and a bar of soap after each test. At baseline testing, participants did not receive food until after the PCL-C instruments were completed. Thus, some may have stayed for the baseline testing even though they could not or would not participate in the study—just to get the food.

With randomization broken, the authors then matched the 21 TM participants on age, sex, and baseline PCL-C scores with 21 of the 51 participants randomized to the delayed-start group. Once the participants learned TM, there was no further attrition from the study. There were 13 men and 8 women in each group. The study was single blind, with the Congolese test administrators who collected the data being blind to group membership. The Ugandan TM teachers and liaisons, who coordinated the activities and knew the group membership, were not involved in data collection or analysis. Half the participants reported meditating twice a day, the rest, at least once a day.

Figure 3 displays that there was a significant 38.7-point decrease in PCL-C scores in the TM group compared to a 6-point increase in the control group (*p* = 0.00000000000001). The effects were similar for women and men. This is the largest reduction in PTSD of all the studies on TM, perhaps because they were the most highly traumatized of any group. Investigation of the individual scores revealed that 90% of the TM participants (19 of 21) endorsed no item more than “a little bit” at both 30- and 135-day posttests. None of the controls showed similar levels of reduction in PTSD at any time during the study. No harmful or adverse effects were spontaneously reported by the experimental group or noted by teachers during the follow-up meetings.

The results of this pilot study are to be viewed with caution because of the small N, the loss of random assignment, and the partial loss of blinding at the 30-day posttest. However, the use of external testers at the 135-day posttest assured it was blind. Our previous meta-analysis of the effects of meditation techniques for treating PTSD [31] reported 61 papers, of which two studies had effects large enough to be considered as outliers. These were Rees et al., 2013 and 2014 [25,52] on the effects of TM on Congolese war refugees with PTSD in Uganda. The effects of g = −5.23 and −2.84 were three to five times larger than the mean effect of all other studies, large enough to distort statistical analyses. We omitted these studies from the main analysis but studied their effects. With these two outliers included, the overall effect of TM was g (SE) −1.31 (0.11), which was reduced to −1.13 (0.09) when the outliers were excluded. The one-study-removed analysis found that removing Rees et al., 2013 [25] only decreased the overall summary effect of TM from −1.31 to −1.19, with a *p*-value that was essentially zero. These effects are clinically meaningful with or without the outlier included and call for larger-scale studies investigating the efficacy of TM practice across cultures.

Study Characteristics. Table 1 shows the characteristics for each of the studies, arranged to show the 7 randomized controlled trials (RCTs) at the top of the table and the 5 controlled trials (CTs) below that. Within the RCT and CT categories, the studies are listed chronologically by publication date. Abbreviations for control groups are AMM = Adapted Mantra Meditation, PTC = Patient Centered Therapy, TAU = Treatment as Usual, WLC = Wait List Control, PE = Prolonged Exposure Therapy, HE = Health Education for PTSD, Facilitated Group Support = group therapy facilitated by an experienced social worker, PT = Psychotherapy, and NT = No Treatment.

Table 1 shows 12 papers that reported using TM to treat PTSD. Of these, three papers reported two comparisons of TM with controls: Heffner et al., 2013 [13] compared TM with PCT and AMM; Nidich et al., 2018 [28] compared TM with PE and HE; and Yoshimura et al., 2015 [38] compared TM with tsunami/earthquake victims in two different cities. 

Table 2 gives the means, standard deviations, and range for the subjects’ age, sample size, % males, and longest study duration from pretest to posttest. It shows that the mean age of the subjects across all studies was 40.5 years, ranging from 20.6 to 54.4. The mean total number of subjects in the studies was 91.8, with a large range of 18 to 238. Across all studies, there were about an equal number of males as females (46.9% males on average), but the studies ranged from all males (100%) to all females (0% males). The mean study duration was 12.3 weeks, with the shortest study being 1 week and the longest 19 weeks.

#### 3.1.2. Repeated Measures Studies on TM: The Rate of Decrease in PTSD Symptoms

In the previous section, the comparisons of TM with control conditions used the final posttest for each study, which varied from 1 to 19 weeks. In Figure 4, we present the results from the five studies that used repeated measures designed to assess how quickly PTSD symptoms decreased through TM and control conditions.

**Figure 4 medicina-61-00659-f004:**
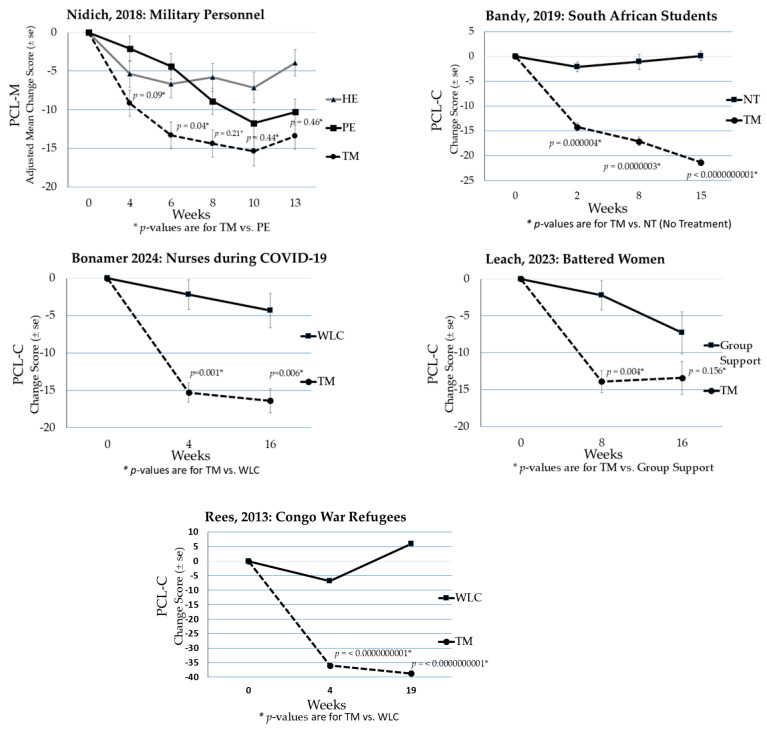
Five studies using repeated measures: Military personnel, South African college students battered women, clinical nurses during COVID, and Congo war refugees. The studies were Nidich et al., 2018 [28], Bandy et al., 2019 [24], Bonamer et al., 2024 [26], Leach and Lorenzon, 2023 [27], and Rees et al., 2013 [25].

Figure 4 shows that in all trauma populations studied, TM practice produced a more rapid reduction in PTSD symptoms than the various comparison treatments. The Nidich et al., 2018 [28] study on veterans found that TM produced a clinically meaningful decrease in PTSD symptoms at weeks 4 and 6 that was statistically larger than the effects of one of the most highly recommended treatments for PTSD, exposure therapy (PE). By week 4, TM reduced the PCL by 9.2 points and by 13.3 points by week 6, whereas PE had only reduced the PCL by 2.0 and 4.3 points over the same periods (TM vs. PE, *p* = 0.09 at week 4, *p* = 0.04 at week 6). This indicates that TM is fast-acting relative to PE. Whereas TM reduced the PCL by 9.2 points in 4 weeks, PE took 8 weeks to reduce symptoms by 8.9 points.

Similarly, Leach and Lorenzon, 2023 found that after 8 weeks at the end of the study on female survivors of interpersonal violence, facilitated Group Support was not statistically inferior to TM, but TM worked faster than Group Support [27]. It can be seen in Figure 4 that after two months (8 weeks), TM reduced PT symptoms by 13.9 points. This reduced level of symptoms was maintained by the TM group after four months (16 weeks), whereas after four months, Group Support had only reduced PTSD symptoms by 7.3 points.

The Bandy et al., 2019 [24] study on South African students with PTSD showed even more rapid symptom reductions. In two weeks, TM reduced the PCL by 14.2 points, which is well below a 10-point reduction, the criteria of a clinically meaningful improvement. By week 15, these students decreased by 21.3 points on the PCL, to a score of 32, which is below the criterion for PTSD diagnosis. The rapid reduction in PTSD in the students may have been influenced by their youth and because they practiced TM twice daily which was built into their college curriculum.

The largest and most rapid reductions in traumatic symptoms were in the Congo refugees [25], who decreased by 36 points on the PCL after 4 weeks of TM and further decreased to below the criterion for PTSD by 19 weeks. In contrast, the wait-list control group showed a mild increase in PTSD symptoms, as life as a refugee remained very stressful. These results support the conclusion that the very large reduction in the initially very high scores did not represent regression to the mean. A similar increase in PTSD over time in untreated war refugees living abroad was found in the Ukrainian refugees control group by Didukh and Freytag, 2024 [47].

### 3.2. Part 2: Meta-Analytic Synthesis of Between-Group Effects

Forest Plot. Meta-analysis was conducted to synthesize the evidence from this set of studies to predict the usefulness of TM as a treatment for PTSD. This was accomplished by pooling results to provide estimates of the effect size, heterogeneity, and effects of covariates. The random-effects model was employed because the studies differed from each other in various ways: active control vs. passive controls, trauma types studied (war, natural disasters, interpersonal violence, sexual abuse), population (war veterans, war refugees, male and female prison inmates, female survivors of IPV, earthquake victims), research design, gender, and age.

Effect Sizes. This forest plot includes two non-inferiority studies comparing TM with established treatments, prolonged exposure, Nidich et al., 2018 [28], and facilitated group support, Leach and Lorenson, 2023 [27]. These non-inferiority studies would be expected to give smaller effect sizes in between group analyses. The rest of the studies were superiority analyses.

The largest effect was the Bandy et al., 2019 [24] study on college students in South Africa comparing TM with no-treatment controls, for an effect size of g = −2.28. The smallest effect was Nidich et al., 2018 [28] comparison of TM with Prolonged Exposure, g = −0.16. The pooled mean effect size of all 14 studies is shown at the bottom of Figure 5, g (95% CI) = −1.01 (−1.29, −0.74). This indicates that the mean effect in the universe of comparable studies could fall anywhere in the range from −1.29 to −0.74 [23,29], all large effects, even when including the non-inferiority studies.

Q, Tau squared, Tau, and Prediction Interval. The Q statistic indicates the amount of dispersion within the various groups under the fixed-effect model. The expected value of Q is the degrees of freedom of Q, which is the number of studies minus 1; in this case, df(Q) = 13. The actual Q-value = 57.71, much larger than the expected value, and *p* = 0.00000013. That is, it is highly likely that the differences observed in the effects across the 14 comparisons of TM with controls were true differences, not just due to random sampling error [23,29].

The I-squared statistic is 77%, which tells us that 77% of the variance in observed effects reflects variance that is true effects rather than sampling error. Tau-squared, the variance of true effect sizes, is 0.241 in g units. Tau, the standard deviation of true effect sizes, is 0.49 in g units. If we assume that the true effects are normally distributed (in g units), we can estimate that the prediction interval is −2.04 to 0.01. The true effect size in 95% of all comparable populations falls in this interval [29], which is basically from a large effect of g = −2.0 to no effect (0.01). The prediction interval says that future studies on TM for treating PTSD compared to controls are likely to produce a range of effects from large reductions in PTSD to no effect. It is important to note that the prediction interval does not predict any studies in which PTSD would increase.

One-study Removed Analysis. Removing one study at a time from a meta-analysis, while including all the other studies quantitatively indicates the influence of each study on the overall summary effect size. It addresses the question of how the conclusion about the effectiveness of the treatment has been influenced by each of the studies. In the present meta-analysis, no individual study reduced the effect size of the summary effect of all studies to less than −1.0 g, a large effect.

Publication Bias: Funnel Plot. One potential source of bias in meta-analysis is that small studies not supporting the researchers’ hypothesis may not be published. This would show up in a funnel plot as more positive studies on one side of the plot than negative studies on the other side. In the present meta-analysis, there was no significant skew in the funnel plot that would suggest bias. Egger’s regression intercept was *p*-value (2-tailed) = 0.93, not significant. Begg and Mazumdar rank correlation was also not significant, both without and with Kendall’s continuity correction (*p*’s = 0.7 and 0.74, respectively). This says that publication bias does not pose a threat to the underlying truth of the synthesized evidence of this meta-analysis.

Fail-Safe N. The Classic fail-safe N analysis found that 822 non-significant studies at the alpha 0.05 level, two-tailed test, would be needed to reduce the significance of the 14 observed studies to non-significance. Orwin’s fail-safe N found that it would take 101 studies with a small effect size of g = −0.2 to reduce the pooled effect of the studies in this meta-analysis to a trivial level of g = −0.3. Thus, it is highly unlikely that there exist low-effect or no-effect missed studies that could nullify or substantially reduce the observed results.

Research Quality. One of the strengths of the studies on TM as treatment for PTSD is that they all used validated measures of PTSD. Additionally, they all used equivalent measures in the pretest and posttest phases, which ensures that the change in PTSD is validly assessed. Another strength in the TM research is the use of skilled personnel to provide the treatment. All TM teachers involved in the reported research studies are certified by the Maharishi Foundation International. In research on meditation, it is important that the studies ensure that the meditation techniques are practiced correctly. 83% of the studies on TM reported systematic follow-up and checking of the correct practice of it. Studies should also ensure that the control group has equivalent systematic follow-up and checking of correct practice of the control or comparison, and on this, most TM studies were lacking. Few (17%) reported systematically checking the efficacy of the control technique.

Implementation Results. We found in all studies that TM was acceptable to the various populations to which it was taught for treating PTSD. Of the subjects to whom TM was offered, 93% learned and 89% completed the course, and 94% participated in post-testing. Half the studies monitored and reported regularity of TM practice. Of the studies reporting, a mean of 75% of the subjects practiced the technique at least once a day.

Meta-Regression. Table 1 and Table 2 show that the studies in this meta-analysis varied on a number of variables, such as age, sample size, % males, study duration, and type of trauma. We used meta-regression to assess the degree to which different covariates explained effect sizes of the studies. The first step was to assess how much each covariate taken separately influenced effect sizes. The second step was to assess which combination of covariates has the most explanatory power.

Table 3 summarizes the regressions of 10 covariates on effect sizes in g units. The table shows the *p*-values and the R^2^s, which are the r^2^ analogs, which is the proportion of variance explained by the covariate, and the I^2^, the variance left unexplained by the covariate. The last column on the right classifies covariates as either research design or subject variables. The covariates are entered in the table according to their R^2^, in descending order. Note that as R^2^ decreases going down the table, that I^2^ increases, as expected.

Research design variables tended to have a greater influence on the effect size than subject variables. TM was effective for subjects of different ages and for military personnel as well as civilians. It worked equally well across patients suffering from different sources of trauma, for males as well as females, irrespective of the person’s level of PTSD at baseline.

On the other hand, research design variables have significant effects. Specifically, studies with stronger control groups had smaller effect sizes. This is because between-group effect sizes are calculated as the effect of TM minus the effect of the control. Consequently, the larger the effect of the control, the smaller the effect size. For example, in the study by Nidich, et al., 2018 [28], TM was compared with a first-line treatment for PTSD, prolonged exposure, and that is the study that had the smallest effect, as seen in Figure 5.

#### Scatter Diagrams

Figure 6, Figure 7, Figure 8, Figure 9, Figure 10 and Figure 11 are the scattergrams for the covariates, which give an intuitive insight into how these covariates work. Each circle in the scatter diagram represents a study, and the size of the circle is inversely proportional to the variance. Larger circles indicate larger, better-controlled studies that carry more weight in the meta-regression.

The strongest covariate was Clear Score, which is a measure of the strength of research design. Clear Score had R^2^ = 0.78, which indicates that it explains 78% of the true between-groups variance, *p* = 0.000007. I^2^ = 42.97, *p* = 0.05, indicating that there was a significant amount of true variance left unexplained. In Figure 6 we see that studies to the left of the chart, which had lower research quality, produced greater reductions in PTSD than studies to the right of the chart that had higher research quality. This illustrates how stronger control groups lower the effect size.

The scatter diagram comparing RCTs and CTs (Figure 7) shows that CTs as a group showed lower between-group effect sizes (greater reductions in PTSD) than RCTs. This is a similar result to Clear Scores, since random assignment is a key element of strong research design.

In Figure 8, we see that ineffective control groups that do not reduce PTSD much and even increase it slightly (to the right) are associated with a larger reduction in PTSD (downward direction of the regression line). Clearly, studies with passive control groups on the left showed greater reduction in PTSD for TM studies than the studies with active control groups on the right (*p* = 0.008, Figure 9). Age of the subjects was a significant covariate (*p* = 0.015), but it only accounted for a small amount of the variance, R^2^ = 0.24 (Figure 10). However, it left almost all of the remaining variance unexplained, I^2^ = 71.71. The scatter diagram shown above shows the significant regression was mostly due to one study, the Bandy et al., 2019 study on South African college students [24], mean age 21 (extreme left circle). The possibility that younger students may get greater benefit from meditation needs to be studied further.

The scatter diagram of military vs. civilian (Figure 11) is interesting because it shows a greater diversity of effects of TM on civilians than on military groups, as well as showing that civilian groups had a larger mean reduction in PTSD. There may be greater diversity of the civilian groups in ethnicities and types of traumas they experienced. Civilian groups included American male and female prisoners, clinical nurse survivors of the COVID-19 pandemic, South African college student survivors of racism and interpersonal violence, Japanese citizen survivors of earthquake, tsunami, and nuclear reactor meltdown, Australian female survivors of interpersonal domestic violence, and Congo and Ukrainian war refugee survivors of the ravages of war. The military groups were also a heterogeneous group with a variety of trauma experiences but were less diverse than the civilians.

## 4. Discussion

Previous meta-analyses had found that treatment of PTSD by meditation techniques reduced PTSD statistically significantly in the range of g = −0.30 to −0.5, which is a moderate clinical effect. Our recent systematic review of all meditation techniques for treating PTSD (61 studies) found an average reduction across all studies of within group effects (posttest minus pretest) effect of g = −0.67. Comparisons among the four types of meditation found that the TM had an effect of g = −1.13, which was a significantly larger effect than the other categories. The present follow-up study was of the between-group effects of TM compared to controls in each study.

We located 15 controlled trials of TM for treating PTSD. The summary within-group effect across all studies was large, −1.01, 95% CI = −1.29 to −0.74. In every study, TM had within-group effects of over −10 points on the PTSD Check List (PCL). The between-group analysis found that TM was superior to psychotherapy, Adapted Mantra Meditation, Patient Centered Therapy, wait-list controls, treatment-as usual, and no treatment. It was not superior to Patient Centered Therapy, Group Support, or Prolonged Exposure therapy (PE). TM was non-inferior to PE at week six. For the five studies that had data on repeated measures for more than one posttest, TM produced clinically meaningful reductions in PTSD by 10 points or more on the PCL within a month or less. The one-study-removed analysis found that no one study reduced the pooled effect of TM to less than g = −1.0. Only Leach and Lorenzon, 2023 [27] reported adverse events. This was a study of female survivors of IPV. Twelve adverse events reported by six TM group participants were transient and mostly mild (i.e., nausea, headache, irritability, and weight gain). Two participants self-reported a severe adverse event that they believed was related to the intervention (i.e., cold sore, body feeling heavy). Five adverse events reported by five participants in the control group were of mild-moderate severity and transient in nature and included shaking, feeling over-whelmed and upset, and heart palpitations. The frequency of adverse events was not shown to be statistically significantly different between groups. The TM subjects expressed that they were “grateful for the experience”, that it was “easy to learn”, that the instructors were “warm and accepting”, that the classes were enjoyable, and that TM “provided a lot of benefits”, including perceived improvements in mental clarity, eyesight, sleep, stress, and coping, all of which are supported by independent research [27]. Overall, participants in the Support Group found that their discussions with other women were “empowering” [27]. Similarly, previous meta-analysis of a variety of meditations for treating PTSD also did not mention adverse events [12], said there were none [15], or said they were uncommon or mild [53,54].

Multiple meta-regression analysis found that experimental design variables (research quality, Clear scores, random assignment, and active vs. passive controls) had statistically significant influence on effect sizes, whereas subject variables (trauma population, % males, and baseline PTSD) did not. Civilians showed larger reductions in PTSD than military groups, perhaps because their exposure to trauma was more episodic and the military’s was more sustained. Civilian groups also showed a greater spread of effect sizes than military groups because there was a greater diversity of their demographics and trauma experiences than for the military. Studies with stronger research designs showed reduced effectiveness, which, on face value, appears to be a threat to the validity of TM. We have suggested that this effect is due to the strength of the control group, not to the weakness of TM. Indeed, our first paper on this meta-analysis showed that within-group effects, which are pretest to posttest changes in TM with no reference to control groups, were consistently large across studies. The clinical effect of TM, how much it reduces PTSD, was consistently high across studies. However, more high-quality studies comparing TM with other effective treatments are needed.

The general interpretation of these results is that the physiological effect of TM is a state that is the opposite of stress and that with repeated practice the state becomes a trait of lower stress [55]. The tradition from which TM derives holds that ultimately, all the physiological and psychological changes associated with the improvement in PTSD symptoms follow from the phenomenon of “transcending”. In this, the subject experiences subtler levels of thought until they transcend thought altogether and reside in the quietest aspect of the mind, a field of inner wakefulness, conscious only of itself, beyond all the heretofore apparently irreconcilable conflicts that had previously afflicted the subject. The rapidity with which symptoms can improve is consistent with the traditional appreciation that even initial brief experiences of oneself as a field of consciousness can deliver from great distress [56].

For example, a previous systematic review and meta-analysis found that during TM, the stress indicators respiratory rate, skin conductance, and plasma lactate decrease more during TM than in controls during ordinary rest [57]. Other studies have found parallel results for cortisol. Outside of TM, the meditators have lower baseline levels of these parameters than controls. The state of low stress during TM has become a trait lasting into activity [58].

Studies have also shown that regular TM practice reduces stress reactivity outside of meditation in the electrodermal [35], cardiovascular [59,60], and hypothalamic-pituitary-thalamic stress systems [61]. It also reduces stress-related conditions co-morbid with PTSD, such as depression [28,32,45,62], insomnia [63], and it has been shown to increase general cognitive abilities [64,65], and sense of self-worth [66,67,68], all of which are needed for recovery from PTSD. These improvements in functionality appear to stem from TM producing general neurophysiological integration in both cortical and midbrain systems. During TM, cortical EEG coherence in alpha1 in the 8 to 10 Hz range increases, which is the hallmark of transcending. Within the span of a year of regular TM practice, the nervous system habituates to maintaining the alpha1 coherence even during focused cognitive activity [69]. Similarly, after three months of TM practice, the connectivity between areas in the parietal lobe that are involved in emotions (the right precuneus and left superior parietal lobe) is correlated with reductions in anxiety and perceived stress [70]. This evidence suggests that the benefits of TM are based upon improved integration of the nervous system.

On the other hand, prolonged exposure and support groups do not have these effects. They work by other mechanisms. Presumably, prolonged exposure works by Pavlovian extinction, and group support works by cognitive restructuring and psychotherapy. Evidence indicates these mechanisms are not as effective for reducing PTSD as transcending is.

There is no change in belief or life style required to practice TM. Within the limitations of this systematic review, the data indicate that TM is effective in the populations studied in the U.S., Australia, Uganda, Japan, South Africa, and Ukraine. People in the U.S. tend to be increasingly culturally aware and flexible, and in this paper, TM has been successfully taught to groups from different U.S. socioeconomic groups, military personnel, male and female prison inmates, and nurses. Over 700 studies on TM have been conducted in 30 countries, and it is taught in over 100 countries. TM has a long tradition of being a part of evidence-based medicine, and demographics and cultural variability have not been much of an issue [58].

Limitations. We noted the limitations mentioned by authors in the descriptions of each of the 15 studies in this review. Sample sizes were as small as 18 subjects, but three studies had more than 200 subjects, and the average sample size was 92. Most studies used power analysis and used an adequate sample size to detect a statistically significant effect, if there were one. Nevertheless, some study authors called for replication with larger sample sizes to test the reliability of TM and to be able to generalize its effects to larger populations.

A more serious limitation was that the short duration of the studies, which averaged only 12 weeks, is too short to assess whether the subjects would continue with TM practice and whether it would have long-term benefits. Also, there were few studies that directly compared TM with other meditation techniques or with first-line treatments for PTSD. Some authors noted that it is often difficult for the new TM meditators to incorporate the recommended 20 min twice-a-day practice into their daily schedule and recommended they consider coordinating with other meditators and their employers to create time and place to meditate together, which would also have the additional benefits of meditating together in a group.

Future Research. More randomized controlled trials directly comparing TM with front-line treatments are needed to assess its general applicability as a non-trauma focused therapy for PTSD. In addition, more research is needed on the potential synergetic effects of combining TM and other techniques with evidence-based practices from integrative medicine, such as diet, exercise, Yoga, and Ayurveda. Our meta-analysis provides strong evidence that TM is an effective non-trauma focused treatment for PTSD that should be taken to the next level, to Phase III studies of high quality RCTs performed on hundreds of subjects at multiple independent sites.

## Figures and Tables

**Figure 1 medicina-61-00659-f001:**
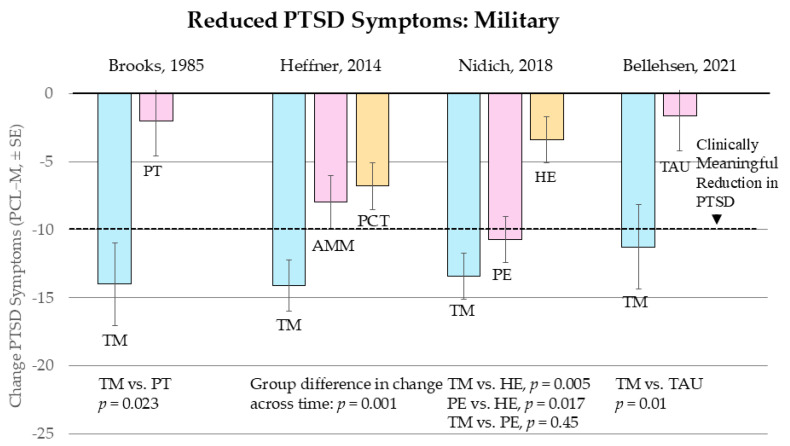
Results of the four controlled trials using the Transcendental Meditation technique for treating PTSD in U.S. military personnel: Brooks and Scarano, 1985 [32]; Heffner et al., 2014 [13]; Nidich et al., 2018 [28]; Bellehsen et al., 2021 [33]. Abbreviations: PT = psychotherapy. AMM = adapted mantra meditation. PE = prolonged exposure. HE = health education. TAU = treatment as usual. In Bellehsen, the change was in the CAPS-5 score.

**Figure 2 medicina-61-00659-f002:**
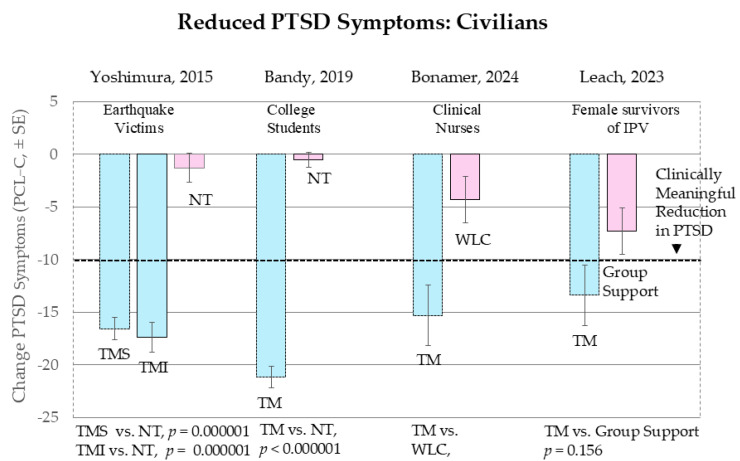
Results of controlled trials using the Transcendental Meditation technique for treating in Earthquake-Tsunami Victims, Yoshimura et al., 2015 [38]; College Students in South Africa, Bandy, et al., 2019 [24]; Clinical Nurses, Bonamer et al., 2024 [26]; and Female Survivors of IPV. Leach and Lorenzon, 2023 [27]. Abbreviations: TMS = TM group in Sendai. TMI = TM group in Ishinomaki. NT = No Treatment. WLC = wait list controls. Group Support = 12 h of facilitated group support, comprising 8 × 1.5-h weekly group sessions delivered over 8 weeks, plus a 1.5-h follow-up session at week 16. As with the four studies on military personnel, these four civilian studies found that treatment with TM produced clinically significant reductions in PTSD symptoms that were greater than those in the control groups.

**Figure 5 medicina-61-00659-f005:**
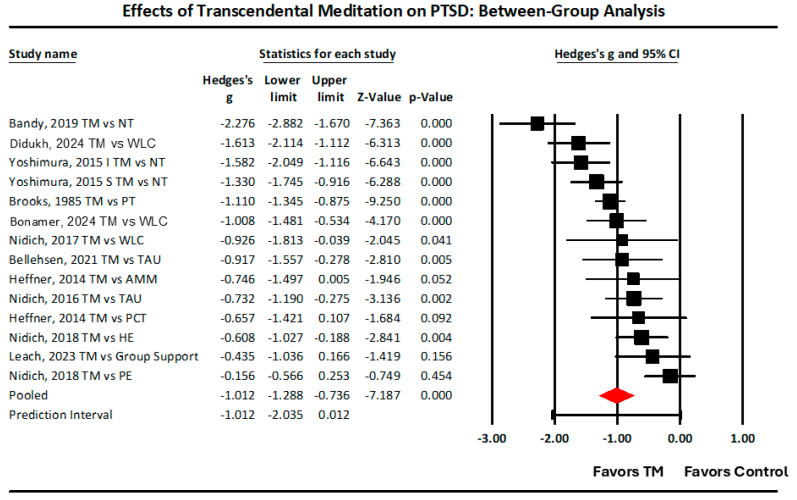
The Forest plot of the effects of 14 independent studies is arranged from the largest to the smallest effect. These studies are Bandy et al., 2019 [24], Didukh and Freytag, 2023 [47], Yoshimura et al., 2015 [38], Brooks and Scarano, 1985 [32], Bonamer et al., 2024 [26], Nidich et al., 2017 [46], Bellehsen et al., 2021 [33], Heffner et al., 2014 [13], Nidich et al., 2016 [45], Nidich et al., 2018 [28], Leach and Lorenson, 2023 [27]. The fifteenth study, Rees et al., 2013 [25], was excluded from this analysis because it was an outlier that would distort to the analysis. The statistics table shows the effect size, Hedges’s g, the lower and upper limit of the 95% CI, the Z-value, and the *p*-value for each study. The areas of the black boxes indicate effect sizes. Their size is inversely proportional to the variances of the study. The red diamond at the bottom indicates the pooled effects of all 14 studies. The center of the diamond marks the mean of all groups (−1.012 g) and its width is the 95% CI (−1.288 to −0.736) Effects less than zero favor TM. Effects greater than zero favor controls. Note that the outlier study (Rees, et al., 2013) [25] is not included in this forest plot because it was out of scale. The control groups are NT = no treatment, WLC = wait list control, PT = psychotherapy, TAU = treatment as usual, AMM = adapted mantra meditation, PCT = patient centered therapy, HE = Health Education for PTSD patients, Group Support = facilitated group support with a social worker, PE = and Prolonged Exposure Therapy.

**Figure 6 medicina-61-00659-f006:**
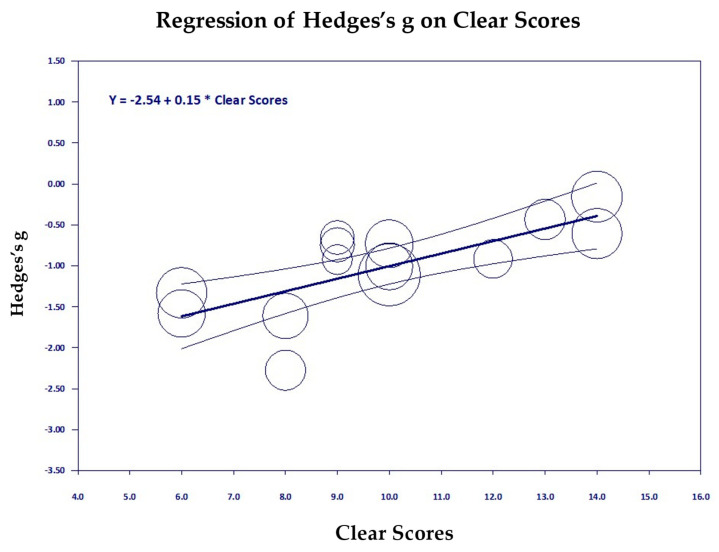
Scatter diagram of Clear Scores, Research Quality. Each circle represents a study. The bold solid line is the linear regression line. The two light lines above and below the linear regression indicate the 95% confidence interval.

**Figure 7 medicina-61-00659-f007:**
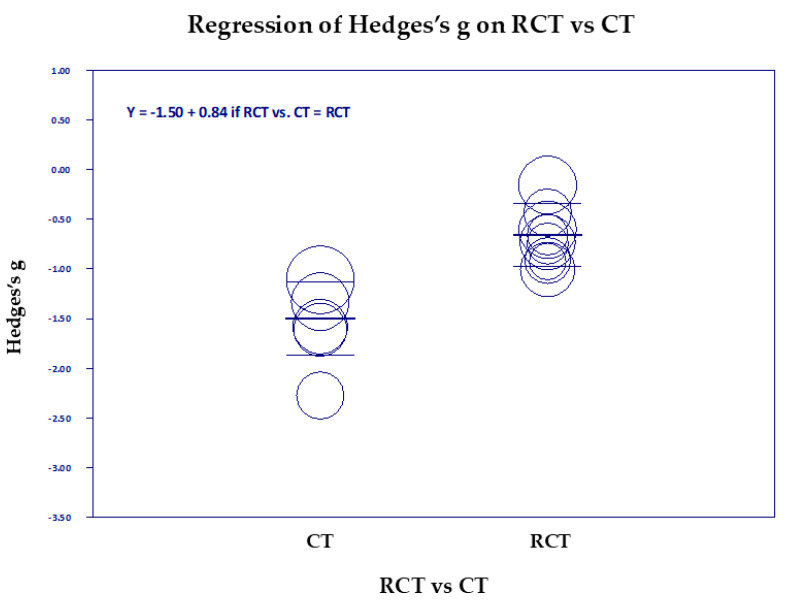
Scatter diagram of RCT vs. CT Each circle represents a study. The bold solid lines are the means of the two groups. The two light lines above and below the means are the 95% confidence intervals.

**Figure 8 medicina-61-00659-f008:**
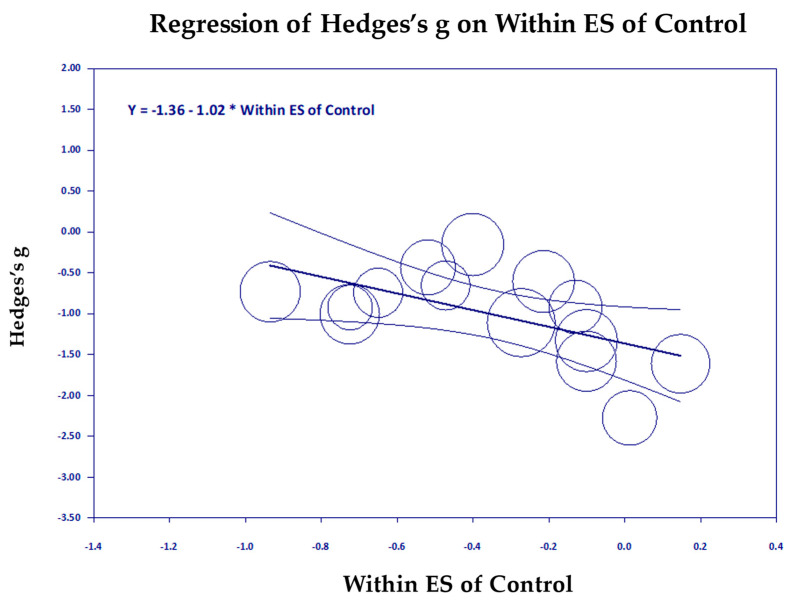
Scatter diagram of Within-Group Effect Size of Controls. Each circle represents a study. The bold solid line is the linear regression line. The two light lines above and below the linear regression indicate the 95% confidence interval.

**Figure 9 medicina-61-00659-f009:**
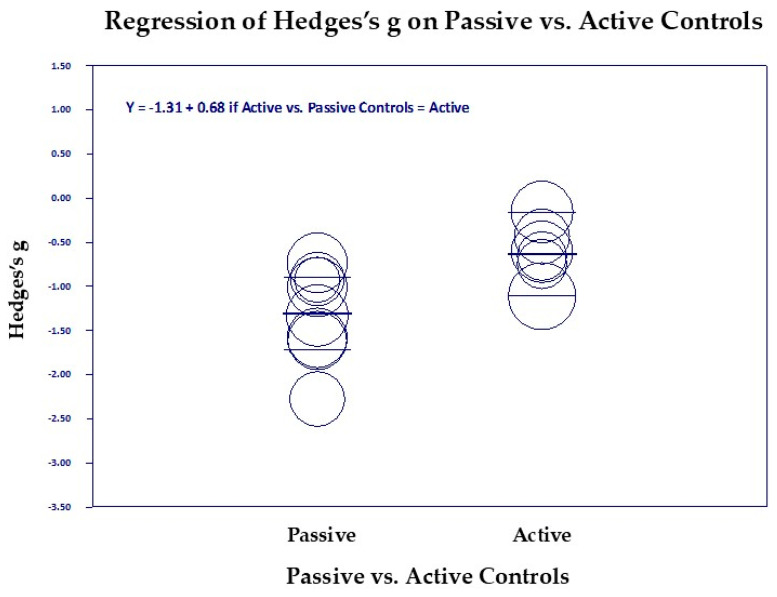
Scatter diagram of Active vs. Passive Controls. Each circle represents a study. The bold solid lines are the means of the two groups. The two light lines above and below the means are the 95% confidence intervals.

**Figure 10 medicina-61-00659-f010:**
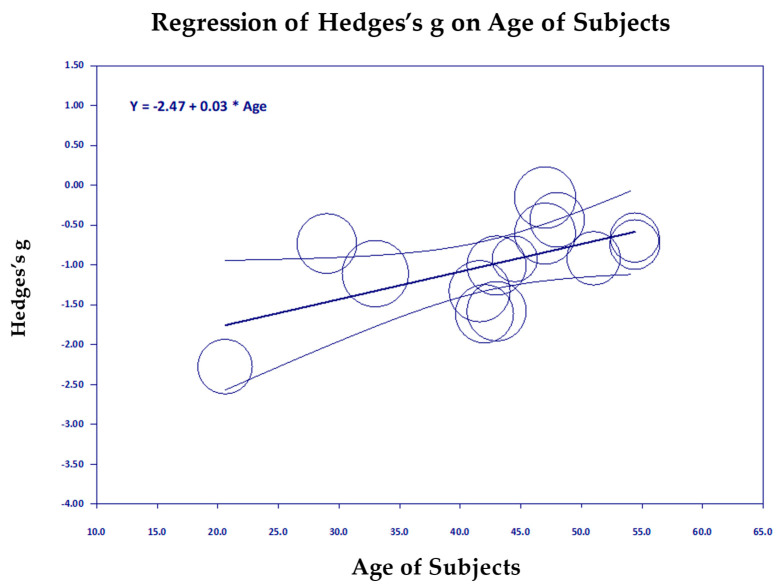
Scatter diagram of Age. Each circle represents a study. The bold solid line is the linear regression line. The two light lines above and below the linear regression indicate the 95% confidence interval.

**Figure 11 medicina-61-00659-f011:**
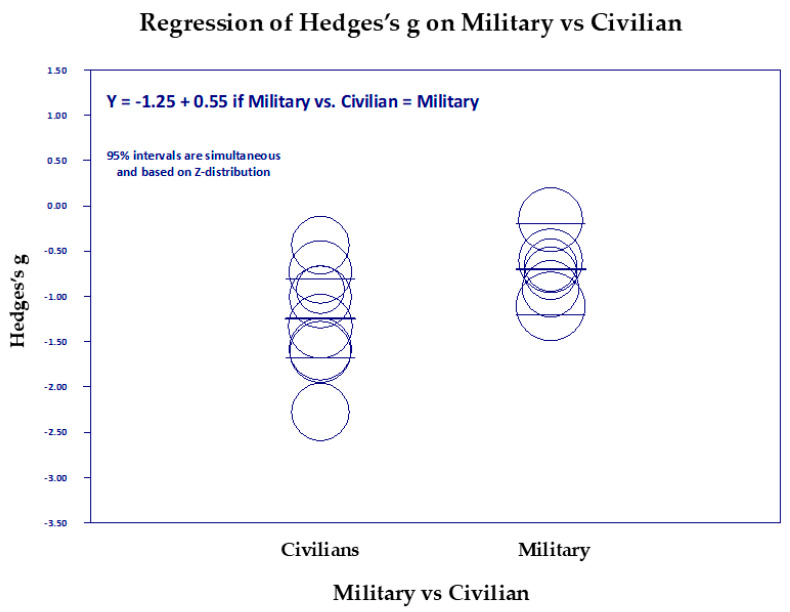
Scatter diagram of Military vs. Civilian. Each circle represents a study. The bold solid lines are the means of the two groups. The two light lines above and below the means are the 95% confidence intervals.

**Table 1 medicina-61-00659-t001:** Characteristics of Studies Using the Transcendental Meditation Technique to Treat PTSD.

Study ID	Location	Population/Trauma Type	Research Design	Cont. Group	Baseline PTSD TM, C	Research Quality	Mean Age TM, C	Sample Total	% Males TM	Study Duration (Week)
Heffner, 2014 Saginaw TM [13]	Saginaw MI VA	Military/Combat-trauma	RCT	AMM, PCT	TM, AMM, PCT 66, 61, 60	7	54, 55	65	74	8
Nidich, 2016 TM [45]	Oregon	Civilian/Male Prison Inmates/Interpersonal violence,	RCT	TAU	35, 31	10	29, 30	181	100	16
	State Penitentiary									
Nidich, 2017 TM [46]	Coffee Creek Correctional Facility, Wilsonville, OR	Civilian/Female Prison Inmates/Interpersonal violence	RCT	WLC	53, 52	9	45, 45	20	0	16
Nidich, 2018 TM [28]	San Diego VA	Military/Combat trauma	RCT	PE, HE	TM, PE, HE 61, 61, 59	14	TM, PE, HE 46, 46, 46	202	82	13
Bellehsen, 2021 TM [33]	Veterans Administration Medical Center Long Island, NY	Military/Combat trauma	RCT	TAU	53, 54	12	53, 50	40	80	12
Leach, 2023 TM [27]	University of South Australia, Adelaide	Civilian/Abused Women, Domestic Violence	RCT	Facilitated Group Support	33, 30	13	48, 47	42	0	16
Bonamer, 2024 TM [26]	Sarasota Memorial Health Care System. Moffitt Cancer Center, Tampa General Hospital	Civilian/Nurses Caregiver Stress	RCT	WLC	29, 25	10	42, 44	104	0	12
Brooks, 1985 TM [32]	VA Denver Colorado, US	Military/Combat-trauma	CT	PT	60, 60	9	33, 33	18	100	12
Rees, 2013 TM [25]	Kampala, Uganda	Civilian/War Refugees, Combat Trauma,	CT	WLC	65, 68	9	33, 31	42	65	19
Yoshimura, 2015 Ishinomaki TM [38]	Sendai, Ishinomaki,	Civilian/Earthquake–Tsunami, Disaster Trauma	CT	NT	I, S, T 33, 34, 23	6	43, 42, 40	239	19	1
	Tokyo, Japan									
Bandy, 2019 TM [24]	Johannesburg, South Africa	Civilian/College Students, sexual and criminal victimization, combat	CT	NT	53, 57	8	21, 21	68	21	15
Didukh, 2024 TM [47]	Lubeck, Germany	Civilian/War Refugees in Ukraine/ Interpersonal violence	CT	WCL	32, 25	8	40, 37	80	23	8

**Table 2 medicina-61-00659-t002:** Summary of Study Characteristics.

	Age of the Subjects	Sample Size N	% Males in the Study	Study Duration (Weeks)
Mean	40.5	91.8	46.9	12.3
SD	2.4	10.1	74.8	40.1
Range	20.6 to 54.4	18 to 239	0% to 100%	1 to 19

**Table 3 medicina-61-00659-t003:** Summary Statistics on the Effects of Covariates: *p*-values of the Covariates, I^2^ and R^2^. (Full details of the meta-regressions can be found in (Appendix A on Basic Analysis and Regression).

Covariate	*p*-Value	R^2^	I^2^	Type of Covariate
Clear Score	0.000007	0.78	42.97, *p* = 0.05	Research Design
RCT vs. CT: CT	0.00002	0.68	51.00, *p* = 0.02	Research Design
Within-Group ES of Controls	0.01	0.34	69.44, *p* = 0.0001	Research Design
Active vs. Passive	0.008	0.26	70.62, *p* = 0.00005	Research Design
Age	0.015	0.24	71.71, *p* = 00003	Subject Variable
Military vs. Civilian	0.04	0.13	73.84, *p* = 0.000007	Subject Variable
Study Duration	0.18	0.13	74.82, *p* = 0.000004	Research Design
Five Trauma Groups	0.2	0.04	76.76, *p* = 0.00003	Subject Variable
% Males	0.18	0	76.63, *p* = 0.0000008	Subject Variable
Baseline PTSD	0.6	0	78.75, *p* = 0.0000001	Subject Variable

## Data Availability

All data used in this study are publicly accessible on the internet from the journal sites that published the papers as indicated in the reference list. In the case of one non-published paper, data is available from the author.

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
