# Peer review of "Systematic Review and Meta-Analysis of Transcendental Meditation for Post-Traumatic Stress Disorder"

_medicina, 2025, doi:10.3390/medicina61040659_

Round 1

Reviewer 1 Report

Comments and Suggestions for Authors

This is a useful clinical preregistered meta-analysis.

Here are, my comments:

- In the abstract clarify the number of study included in the meta-analysis.

- Clarify the exclusion of Rees et al, 2013 study from the overall analysis.

- Discuss why TM effects are almost comparable with prolonged exposure, Nidich et al., 2018, and facilitated group support, Leach and Lorenson, 2023.

- Show the results of the one-study removed analysis in an appendix o supplementary material.

- Similarly show the funnel plot.

- To better understand the clinical effects, we suggest estimating the PCS corresponding to the standardized effect size of the overall meta-analysis results.

- In the Limitations paragraph, consider the relevance of long-term follow-up controls.

- Appendix A, was not available for the review. We recommend including the PRISMA flowchart.

- Following open-science practice, we recommend allowing the paper's reviewers and readers to inspect the raw data used for all calculations for independent controls.

Minor comments:

- In the reference list identify the references related to the studies included in the meta-analysis.

Author Response

Comments and Suggestions for Authors:

This is a useful clinical preregistered meta-analysis.

DOJ. Thank you, for reviewing our study. Replying to our feedback will make it better.

Here are, my comments:

- In the abstract clarify the number of study included in the meta-analysis.

DOJ. It is in our abstract. In the abstract we said: “Searches of data bases (PubMed, MEDLINE, PsycINFO, Web of Science, Library of Congress, and Google Scholar) and bibliographies located 15 controlled studies comparing TM with control groups with 1248 subjects that met our inclusion criteria of being longitudinal, reporting sufficient statistics to calculate effect sizes, all military and civilian populations.”

- Clarify the exclusion of Rees et al, 2013 study from the overall analysis.

DOJ. In the paper we wrote that Rees et al. was an outlier because of the unprecedented large effect size. For that reason it was not included in the meta-analysis, and we speculated why the effect sizes might be so large. Here is what we had said:On p. 12. “The changes in PCL-C scores in this study, 36-point reduction in PCL and the reduction of g = -8.27 (-10.14, -6.41) were larger than any other TM study in this meta-analysis and other interventions on PTSD reported in the literature18. This size of change may have been due to experimental-demand effects in the TM participants to please the teachers, in the hopes that they may help get them into a better situation. Such large effects show the study finding is an outlier. Even so, the clinically meaningful effect it found are consistent with those of the other studies and call for larger scale studies investigating the efficacy of TM practice across cultures.

On p. 14. “Note that the outlier study (Rees, et al., 2013)54 is not included in this forest plot, because it was out of scale.”

In the revised paper, we are attaching the tables and figures of results for individual studies and meta-analytic synthesis and meta-regression. This is “Supplementary Materials Tables S1-S18 and Figures S1-S16 on Basic Analysis and Regression”. It has the effect sizes (in Hedges’s g), SEs, SDs, 95% CIs, and p’s for all studies, with associated summary effects (Random Effects model and Fixed Effect model), and heterogeneity statistics for 15 studies with Rees, et al. 2013 and 14 studies excluding Rees et al. This shows the extent that Rees et al. was an outlier (ES was -8.27), almost 8 times the summary effect of all studies (-1.21). The analysis without Rees et al. only reduced the summary effect to -1.01. But Rees et al. was excluded because it also distorted correlations needed to do meta-regression. Table of results and scatter diagrams of all meta-regression are included in the supplementary materials file, along with a table of definitions of the meta-regression parameters used a guide to interpreting the tables. 

- Discuss why TM effects are almost comparable with prolonged exposure, Nidich et al., 2018, and facilitated group support, Leach and Lorenson, 2023.

DOJ. We added text on p. 16 discussing the different mechanisms of TM, prolonged exposure, and support groups that might explain their differences in effectiveness.

“For example, a previous systematic review and meta-analyses found that during TM the stress indicators respiratory rate, skin conductance, and plasma lactate decrease more during TM than in controls during ordinary rest. Other studies have found parallel results for cortisol. Outside of TM the meditators have lower levels of these parameters at baseline than controls. The state of low stress has become a trait of low stress. Related to this issue, regular TM practice reduces stress reactivity in the electrodermal, cardiovascular systems, and HPA stress system.  TM also reduces stress-related conditions co-morbid with PTSD, such as depression, insomnia, and it has been shown to increase general cognitive abilities, and sense of self-worth. TM also produces state and trait increases in EEG coherence and cortical connectivity that are correlated with decreased anxiety and perceived stress and increased creativity, intelligence, and higher states of consciousness, all conductive to recovery from PTSD.

On the other hand, prolonged exposure and support groups do not have these effects. They work by other mechanisms. Presumably, prolonged exposure works by Pavlovian extinction and group support works by cognitive restructuring and psychotherapy. [Evidence indicates these mechanisms are not as effective for reducing PTSD as TM is.” 

- Show the results of the one-study removed analysis in an appendix or supplementary material.

DOJ. We now included the one-study removed analyses in the Supplementary materials file, which is  

- Similarly show the funnel plot.

DOJ. The funnel plot is shown in the Supplementary materials file that we added.

- To better understand the clinical effects, we suggest estimating the PCS corresponding to the standardized effect size of the overall meta-analysis results.

DOJ. We request clarification and definition of  PCS.

- In the Limitations paragraph, consider the relevance of long-term follow-up controls.

DOJ. We did that, p. 16.

- Appendix A, was not available for the review. We recommend including the PRISMA flowchart.

DOJ. The PRISMA flowchart is in the Supplementary Materials.

- Following open-science practice, we recommend allowing the paper's reviewers and readers to inspect the raw data used for all calculations for independent controls.

DOJ. The Supplementary materials has all the calculations of the primary data, which is in the cited papers.

Minor comments:

- In the reference list identify the references related to the studies included in the meta-analysis.

DOJ. We used the standard reference list style used in scientific papers. We used numbered references, which are arranged sequentially in reference list at the end of the paper, easy to look up.

Reviewer 2 Report

Comments and Suggestions for Authors

Major Comments

Abstract

1. Authors should include the quantitative data to claim “TM is non-inferior to other treatments” with respect to effect sizes, response times. Funnel plots alone cannot eliminate bias concerns; Authors are suggested to clarify or revise it.

Introduction

2. Authors are suggested to add concise explanation with recent literature of current treatments fail for PTSD. Furthermore, authors should explain how psychological mechanisms of TM address PTSD symptoms.

Methods

3. The search methodology spans over five decades (1970–2024), but the inclusion of studies over such a long time period without addressing potential changes in diagnostic criteria and PTSD measures may introduce inconsistencies. Authors are suggested to add inclusion and exclusion criteria. The authors should also explain how the exclusion impacts the generalizability of results.

Results

4. Results are poorly written and explained. Authors are suggested to improve this section by conducting subgroup analyses on population characteristics.

Discussion:

5. Authors claimed that TM is effective for all populations overly broad and not supported by the data. Authors should address the adverse events on the basis of severity and frequency.

6. Authors failed to critically compare the results with similar studies on PTSD treatments. Are the effect sizes for TM comparable to other interventions like mindfulness or EMDR?

7. Authors should discuss the limitations of the study with respect to the potential impact of cultural and demographic variability on the effectiveness of TM.

General Comments:

1. Authors should add abbreviation list used in the manuscript.

2. Proofread the manuscript for language and grammatical issues.

3. Ensure consistency in terminology and formatting throughout the manuscript. Some headings are in bold format and some are not.  

Author Response

Comments and Suggestions for Authors

Major Comments

Abstract

  1. Authors should include the quantitative data to claim “TM is non-inferior to other treatments” with respect to effect sizes, response times. Funnel plots alone cannot eliminate bias concerns; Authors are suggested to clarify or revise it.

DOJ. What we have already said in our Abstract is: “Compared to other first line treatments for PTSD (prolonged exposure therapy and facilitated support groups), TM was non-inferior, and it worked more rapidly. On page 6 in the text we report the results from the paper : “The non-inferiority analysis comparing TM with Prolonged Exposure Therapy (PE) found that TM was non-inferior to PE (p = .0001) on the CAPS score and non-inferior to PE for the PCL-M and PHQ-9 scores (p’s = 0.0002) when covarying using baseline dependent score, number of PTSD medications, gender, and number of years since discharge from the armed forces as covariates. Similar results were obtained when including the following additional covariates: antidepressants and antipsychotic medications at baseline, change in number of PTSD medications, baseline social support, and number of treatment sessions.”

We added this text in the methods section. “Funnel plot analysis found no risk of bias.  Compared to other first line treatments for PTSD (prolonged exposure therapy and facilitated support groups), TM was non-inferior, and it worked more rapidly. The non-inferiority analysis comparing TM with Prolonged Exposure Therapy (PE) found that TM was non-inferior to PE (p = .0001) on the CAPS score and non-inferior to PE for the PCL-M and PHQ-9 scores (p’s = 0.0002) when covarying using baseline dependent score, number of PTSD medications, gender, and number of years since discharge from the armed forces as covariates. Similar results were obtained when including additional covariates: antidepressants and antipsychotic medications at baseline, change in number of PTSD medications, baseline social support, and number of treatment sessions.”

Introduction

  1. Authors are suggested to add concise explanation with recent literature of current treatments fail for PTSD. Furthermore, authors should explain how psychological mechanisms of TM address PTSD symptoms.

DOJ. We added this text in blue to the introduction,

Meditation for Treating PTSD. A variety of essentially non-trauma focused meditation practices have been considered in the treatment of PTSD12-16. In our PROSPERO protocol17, we published our plan to do an updated systematic review and meta-analysis of all treatments identified as “meditation”, all research designs, all populations. Our first paper on this meta-analysis presented the within-group results18, which compared the change from pretest to posttest produced by each of the four meditation categories. We located 61 studies with 3,440 subjects total on Mindfulness-Based Stress Reduction (MBSR, 13 studies); Mindfulness-Based Other techniques (MBO, 16 studies), Transcendental Meditation (TM, 18 studies) and Other Meditations that were neither mindfulness nor TM (OM, 14 studies). The baseline characteristics of subjects were similar across the meditation categories: mean age = 52.2 years, sample size = 55.4, % males = 65.1%, maximum study duration = 13.2 weeks. We found that many different populations with PTSD are willing to try and practice meditation, men and women, young adults and the elderly, war veterans, survivors of Military Sexual Trauma (MST), war refugees, earthquake survivors, prison inmates, survivors of interpersonal violence, and clinical nurses. No study reported serious side effects. There were no significant differences between meditation categories on strength of research design nor evidence of publication bias. Nor were there appreciable differences in types of study populations included, outcome measures, control conditions, gender, or length of the studies.

All meditation techniques had statistically significant effects on reducing PTSD. The strength of the effects in Hedges’s g were moderate for MBSR (-.52), MBO (-0.66), OM (-0.63) and large for TM (-1.13). TM’s effect was significantly larger than for each of the other categories, which did not differ from each other. Meta-regression found that type of meditation was the most significant covariate, accounting for 41% of the variance (R2 = .41), with age accounting for 35%, and trauma groups 26%. Multivariate meta-regression that included these three covariates taken together increased the R2 to .64, 64% of the variance explained by the covariates. 

Having compared the four different meditation techniques in our first paper, the objective of this second systematic review and meta-analysis is to examine the between-group effects of TM, which compares TM with various control groups on the study level. The between group analyses reported compare TM with control groups that are designed to answer such questions as can the effects of TM on PTSD be explained by psychological variables (placebos) such as extra attention they get from just being in the study or explained by expectations of benefits they have from hearing about research on it. Another set of research questions addressed by these between-group studies revolves around how TM compares with well-known current best practices to treating PTSD, such as prolonged exposure or group psychotherapy? Is TM non-inferior to prolonged exposure or group psychotherapy with a professional facilitator?  This paper addresses such questions.

In Part I of this paper, we present each study individually, describing the background of the problem of PTSD in that population, where the study was done, who did it, and present the results and limitations with suggestions for future research that the study authors noted. In Part II we present results of our meta-analysis and meta-regression, including summary effect sizes, 95% confidence intervals, funnel plots and statistics, bias analyses, time course of the effects, and meta-regression findings on covariates that may potentially influence the results, such as study population, type or trauma, the prediction interval (which measures heterogeneity, the range of effect sizes that future studies would be expected to find), the proportion of variance not explained by TM and the control groups or the co-variates (I2), and proposed future research.

  1.  

Methods

  1. The search methodology spans over five decades (1970–2024), but the inclusion of studies over such a long time period without addressing potential changes in diagnostic criteria and PTSD measures may introduce inconsistencies. Authors are suggested to add inclusion and exclusion criteria. The authors should also explain how the exclusion impacts the generalizability of results.

DOJ. We did not exclude any studies on TM and PTSD. Our inclusion criteria are stated on page 4 of our paper. They were very broad, including RCTs as well as CTs on TM and PTSD, and they are up to date, resulting in a greater number of controlled studies on TM and PTSD as previous reviews. Only studies that did not report enough data to calculate and effect size were excluded.

  It is true that there was no PCL criteria scale in 1985 when the first PTSD study on TM was published, but the study measured a broad range of variables that cover PTSD. On. p. 6 we wrote “After 12 weeks there were significant reductions in the TM group compared to psycho-therapy on PTSD symptoms, emotional numbness, depression, anxiety, alcohol consumption, and electrophysiological stress reactivity (skin resistance startle response to loud tone38) as well as improvements in sleep quality, family life and employment problems.” We edited that to include that the PTSD scale used was based on DMS III criteria. I don’t see how the definition of PTSD in this early study differed in any way that could alter the major conclusion of the paper, and all other studies in the review used more modern standard scales of the PCL.

Results

  1. Results are poorly written and explained. Authors are suggested to improve this section by conducting subgroup analyses on population characteristics.

DOJ. Thankyou. We have worked on clarifying the writing.

We added a section on pp 21–28 under Meta-regression on subgroup analysis.

Discussion:

  1. Authors claimed that TM is effective for all populations overly broad and not supported by the data. Authors should address the adverse events on the basis of severity and frequency.

DOJ. The data show that for the broad range of types of PTSD that have been studied with TM that TM is effective. Almost no adverse events were reported, so there is no data on which to study their severity or frequency. We also called for larger, better controlled studies in the future.

  1. Authors failed to critically compare the results with similar studies on PTSD treatments. Are the effect sizes for TM comparable to other interventions like mindfulness or EMDR?

DOJ. Thank you. We added on pp. 2-3 in the Introduction a summary of our first paper in this meta-analysis on 61 studies, which gives the results of comparing TM with mindfulness and other types of meditations.  Previous meta-analyses have not made these comparisons. In Part I of this paper, the results on the study-level that compare TM with a variety of active and passive controls are presented. In Part II on pp. 21 28, we present the results on the meta-analytic and regression-analysis levels for passive and active controls and for several other dimensions.

  1. Authors should discuss the limitations of the study with respect to the potential impact of cultural and demographic variability on the effectiveness of TM.

DOJ. In added text (p. 27 to 28 to cover these issues.  

“The understanding from the tradition from which TM derives is that ultimately, all the physiological and psychological changes associated with the improvement in PTSD symptoms follow from the phenomenon of “transcending.” In this, the subject experiences subtler levels of thought until he transcends thought altogether and resides in the quietest aspect of his mind, a field of inner wakefulness, conscious only of itself, beyond all the heretofore apparently irreconcilable conflicts which had previously afflicted him. The rapidity with which symptoms can improve are consistent with the traditional appreciation that even initial brief experiences of pure consciousness can deliver from distress. [NEED REFERENCES]

For example, a previous systematic review and meta-analyses found that during TM the stress indicators,  respiratory rate, skin conductance, and plasma lactate decrease more during TM than in controls during ordinary   rest. [NEED REFERENCES] Other studies have found parallel results for cortisol. Outside of TM the meditators have lower baseline levels of these parameters than controls. [NEED REFERENCES] The state of low stress during TM has become a trait lasting into activity.

Studies have also shown that regular TM practice reduces stress reactivity outside of meditation in the electrodermal, cardiovascular, and HPA stress system. [NEED REFERENCES] It alsoreduces -stress-related conditions co-morbid with PTSD, such as depression, insomnia, and it has been shown to increase general cognitive abilities, and sense of self-worth, all of which are needed for recovery from PTSD. [NEED REFERENCES] These improvements in functionality appear to stem from TM producing general neurophysiological integration in both cortical and midbrain systems. [NEED REFERENCES] During TM, cortical EEG coherence in alpha1 in the 8 to 10 Hz range increases, which is the hallmark of transcending. [NEED REFERENCES]   Within the span of a year of regular TM practice, the nervous system habituates to maintain the alpha1 coherence even during focused cognitive activity. [NEED REFERENCES] Similarly, after three months of TM practice the connectivity between areas in the parietal lobe that are involved in emotions (the right precuneus and left superior parietal lobe) are correlated with reductions in anxiety and perceived stress. [NEED REFERENCES] This evidence indicates that the benefits of TM are based upon improved integration of the nervous system.

On the other hand, prolonged exposure and support groups do not have these effects. They work by other mechanisms. Presumably, prolonged exposure works by Pavlovian extinction and group support works by cognitive restructuring and psychotherapy. [NEED REFERENCES] Evidence indicates these mechanisms are not as effective for reducing PTSD as transcending is.

There is no change in belief or life-style required to practice TM. Across all studies in this systematic review, the data indicate that TM is effective in populations in the U.S., Australians, Ugandans, Japanese, South Africans, and Ukrainians. People in the U.S. tend to be increasingly culturally aware and flexible, and in this paper TM has been taught to groups from different U.S. socioeconomic groups, military personnel, male and female prison inmates, and nurses. Over 700 studies on TM have been conducted in 30 countries and it is taught in over 100 countries. TM has a long tradition of being a part of evidence-based medicine and demographics and cultural variability have not been much of an issue.”

General Comments:

  1. Authors should add abbreviation list used in the manuscript.

DOJ. We added these.

  1. Proofread the manuscript for language and grammatical issues.

DOJ. We did this.

  1. Ensure consistency in terminology and formatting throughout the manuscript. Some headings are in bold format and some are not.

DOJ. We did this.

Reviewer 3 Report

Comments and Suggestions for Authors

The subject of this review is an important topic that highlights the significance of Transcendental Meditation in treating PTSD. However, the scientific tone is lacking, and many sentences are difficult to follow. Therefore, I suggest a critical revision to improve clarity, refine several sentences, and correct the citations."

A few references are inappropriate and do not match the text and reference list.

For example, on Page 3, paragraph with the heading Dependent variable, in the last line of the first paragraph, the authors mentioned the name Dr Maxwell Rainforth with reference number 31, whereas, in the reference list, the given number cited different authors (Nidich S et al., 2018).

Similarly, in the next heading, "Calculating effect size," the authors mentioned that "our methods followed an introduction to meta-analysis" (citation number 27), whereas, in the reference list, article number 27, they are different authors from the current manuscripts.

Comments on the Quality of English Language

Scientific tone is lacking, and many sentences are difficult to follow. 

Author Response

Comments and Suggestions for Authors

The subject of this review is an important topic that highlights the significance of Transcendental Meditation in treating PTSD. However, the scientific tone is lacking, and many sentences are difficult to follow. Therefore, I suggest a critical revision to improve clarity, refine several sentences, and correct the citations."

DOJ. Thank you for your encouraging perspective and these  guidelines. We revised the paper according to them. Please see text in blue font in the paper.

A few references are inappropriate and do not match the text and reference list.

DOJ. Thank you. We have double-checked all the references.

For example, on Page 3, paragraph with the heading Dependent variable, in the last line of the first paragraph, the authors mentioned the name Dr Maxwell Rainforth with reference number 31, whereas, in the reference list, the given number cited different authors (Nidich S et al., 2018).

DOJ. Dr. Rainforth was the statistician on the Nidich. 2018 paper and is in the reference  citation. 

Similarly, in the next heading, "Calculating effect size," the authors mentioned that "our methods followed an introduction to meta-analysis" (citation number 27), whereas, in the reference list, article number 27, they are different authors from the current manuscripts.

DOJ. Reference 27 is the correct citation here.

“27.        Borenstein, M.;  Hedges, L.;  Higgins, J.; Rothstein, H., Introduction to Meta-Analysis (Second ed.). John Wiley & Sons, Ltd: Southern Gate, Chichester, UK, 2021.”

Round 2

Reviewer 2 Report

Comments and Suggestions for Authors

Authors have diligently addressed almost all the comments and concerns raised during the review process. The revisions made have significantly improved the quality of the article. However, authors must re-evaluate the content of the article, ensuring that the similarity index falls within the accepted limit (below 20%).

Author Response

Thank you for the appreciation of our improvements. Here we will reply to the issue about the Similarity Index. A similarity index is a percentage that measures how much of a text matches other sources. Generally that would mean to compare our study with the results of similar previous studies. However, in the case of our paper, there have not previously been any systematic reviews and meta-analyses on the effects of TM on PTSD.  So, we can compare how closely the data in our review matches the data reported in the original research papers on TM for treating PTSD. What we have done is to compare the data from the tables in the original research papers with the data that has been put into our meta-analysis program and to calculate a % similarity. We have done that and find that it is 100%. Below is the documentation. Below is a forrest plot showng the results we reported in the paper. To the left of that is a column showing the effect sizes for each study in Hedges's g, next to a column of the numbers from the meta-analysis program shown below. Below that you will see copies of the table that the data came from and below that the Summary table of inputs and results calculated by the CMA program, version 4, which show rows of data for each study from the meta-analysis program and the calculated ES Papers. You can see that the data entry from the research papers into the meta-analysis progam is 100% accurate.

Reviewer 3 Report

Comments and Suggestions for Authors

The manuscript seems improved, however, I still think the scientific tone is lacking. I have no further comments.

Author Response

See below for examples of technical language. We corrected all the references and checked them against the text. To our knowledge they are now correct.

----------

Examples of the use of technical language in our manuscript medicina 3381954 (4) DOJ.doc

This document quotes from our paper some examples of our use of appropriate technical language and methodologies on research on PTSD. Technical language is in italics.

Introduction. Problem. We describe the symptoms of PTSD: “dissociative reactions (flashbacks) and nightmares; persistent avoidance of stimuli and memories associated with the trauma; uncontrollable intrusive thoughts and mood swings; and marked alterations in arousal, such as panic attacks, depression, and insomnia.”

We gave statistics on prevalence pf PTSD.

We reviewed the limitation of current treatment for PTSD that used “pharmaceuticals and/or forms of trauma-focused cognitive behavioral therapy, such as exposure therapy.”

Our review used technical terminology of the fields. “Meditation for Treating PTSD. A variety of essentially non-trauma focused meditation practices have been considered in the treatment of PTSD”

We cited our PROSPERO protocol for this systematic review, which specifies what we did on technical detail.

We summarized the results of our previous paper on all types of meditation for treating PTSD, which found TM most effective. We gave the structure of the present paper in terms of the technical terms used in systematic reviews. “In Part I of this paper, we present each study individually, describing the background of the problem of PTSD in that population, where the study was done, who did it, and presenting the results and limitations with suggestions for future research that the study authors noted. In Part II we present results of our meta-analysis and meta-regression, including summary effect sizes, 95% confidence intervals, funnel plots and statistics, bias analyses, time course of the effects, and meta-regression findings on covariates that may potentially influence the results, such as study population, type of trauma, the prediction interval (which measures heterogeneity, the range of effect sizes that future studies would be expected to find), the proportion of variance not explained by TM and the control groups or the co-variates (I2), and proposed future research.

In Methods, we technically described the searches: “Following Preferred Reporting Items for Systematic reviews and Meta-Analyses guidelines (Prisma, https://www.prisma.io/), we searched major databases (PubMed, MEDLINE, PsycINFO, Web of Science, Library of Congress, and Google Scholar), as well as the bibliographies of systematic reviews and meta-analyses, bibliographies of original research papers, and research anthologies and data bases of meditation research, from January, 1970 through June 2024, for papers in English, published and unpublished, on Transcendental Meditation, TM and PTSD. We searched the bibliographies of major reviews and meta-analyses on meditation research.”

We described the inclusion and exclusion criteria: “Inclusion criteria were populations at least presumptively diagnosed with PTSD that had TM in at least one arm of the study, all age groups, and all gender identifications. Populations included combat veterans, war refugees, male and female prison inmates, college students, urban trauma survivors, and nurses under stress.”

Studies were excluded if they did not have sufficient data for calculating effect sizes, not longitudinal with a pretest and at least one post-test, were correlational and did not measure treatment effects, were reviews only, and studies where we did not have permission to use their data.”

Calculating Effect Size. We described how sizes were calculated in technical terms. “Our methods followed Introduction to Meta-Analysis. The random-effects model was used because of the wide differences between study populations on initial levels of anxiety, age, and other variables. The core analysis included the point estimate of the standard difference in the means, Hedges’s g, including standard error, variance, 95% confidence interval (95% CI), Z score, statistical significance, and Forest plots. Heterogeneity was assessed by using the Q statistic and its associated p-value, and the I2. I2 indexes heterogeneity by giving the proportion of the observed variance that is real between-studies variance, in contrast to variance due to sampling error of a common underlying effect (fixed effect).

We described in technical terms how outliers were defined. “Outlier. The Rees, 2013 study was an outlier, with an ES of g = -8.273, which was almost 8 times the pooled ES of all 15 studies taken together, g = -1.208. Removing Rees, 2013 only reduced the pooled effect from -1.208 to -1.01. One-study removed analysis found that removal of Rees resulted in a point value of -.9895, and the point-value of removing any of the other studies was over -1.0. (See Supplementary Materials Tables S1-S2 and Figures S1-S4 for details).

Synthesis Methods, Three-Armed Studies, and Bias Analysis also were in the technical language of systematic reviews and meta-analyses.

Results. In Part 1, in the section called Between Group Analyses Comparing TM with Various Controls we presented the details of where the study was done, the inclusion criteria, PTSD measure, description of the control/comparison group, random assignment (or other) procedure, magnitude or the results, statistical significance, clinical relevance, and limitations and suggested future research.

Etc.

We conclude that there was sufficient technical and scientific language to describe the various sections of the research. If the reviewer would like to suggest areas that were not acceptable we would be happy to revisit them.”

Round 3

Reviewer 3 Report

Comments and Suggestions for Authors

A few sentences in the manuscript could be more precise; for example, these sentences in the abstract-

Previous meta-analyses have found that meditation techniques are beneficial for reducing PTSD,...

We found TM produces large, rapid, clinically significant reductions in PTSD in all groups studied— 

Reducing PTSD sounds vague.

References that are in preparation could be removed, such as ref number 52. 

Comments on the Quality of English Language

As I mentioned in my comments, a few sentences sound vague and could be more precise. 

Author Response

Dear Editor,

Please find the latest revision of our paper, Systematic Review and Meta-Analysis of Transcendental Meditation for Post-Traumatic Stress Disorder, Medicina-3381954 (19) 3.10.2025 uploaded to your website as submission for publication. This revision is a point-by-point response to concerns of Reviewer 3, as documented below. We also did a thorough proof reading of the whole paper and our edits are indicated by blue font color.

  Reviewer 3. “A few sentences in the manuscript could be more precise; for example, these sentences in the abstract-

“Previous meta-analyses have found that meditation techniques are beneficial for reducing PTSD,...

We found TM produces large, rapid, clinically significant reductions in PTSD in all groups studied— 

Reducing PTSD sounds vague.

Authors reply: Thank you for these specific examples. It makes it clear that you want quantitative summaries of the findings in the Abstract and throughout the paper. We rewrote the Abstract and edited the entire paper for changes to explain things more clearly. The abstract is now a little long to comply with these suggestions. We hope this is acceptable.

 Reviewer #3: References that are in preparation could be removed, such as ref number 52. 

Authors reply: Reference #52 is not a “reference in preparation”. It is to a Wikipedia article: “52. Wikipedia authors Second Congo War. https://en.wikipedia.org/wiki/Second_Congo_War.

” In a previous round of replies we added a second reference to the Congo War, “53. Center for Preventive Action Conflict in the Democratic Republic of Congo. https://www.cfr.org/global-conflict-tracker/conflict/violence-democratic-republic-congo.”

Reference 48 is to the paper by Didukh and it is the only manuscript in preparation in our reference list.: “48. Didukh, M. L.; Freytag, K. S., Reduction in symptoms of posttraumatic stress disorder and depression in Ukrainian refugees practicing Transcendental Meditation. in preparation 2024.”

We included Didukh because in our published protocol we said: “ Searches. Following Preferred Reporting Items for Systematic Reviews and Meta-Analyses (Prisma, https://www.prisma.io/) guidelines, we will search for papers in English and all other languages, published and unpublished, on the Transcendental Meditation technique for treating PTSD.”

Our rationale is because including unpublished papers in a meta-analysis is crucial to mitigate publication bias, which occurs when studies with positive results are more likely to be published, leading to an incomplete picture of the evidence1. Moreover, one-study removed analysis showed that including the Didukh paper made virtually no difference in the summary effect. That is, including Didukh did not inflate the overall effect size of all the studies taken together.

  1. Borenstein, M.; Hedges, L.;  Higgins, J.; Rothstein, H., Introduction to Meta-Analysis (Second ed.). John Wiley & Sons, Ltd: Southern Gate, Chichester, UK, 2021.

Yours sincerely,

David W. Orme-Johnson, PhD